# Diverse Condensed Data Generation via Class Preserving Distribution Matching

**Dandan Guo**                                                      *guodandan@jlu.edu.cn*
*School of Artificial Intelligence*
*Jilin University*

**Zhuo Li**                                                      *zhuoli3@link.cuhk.edu.cn*
*Shenzhen International Center for Industrial and Applied Mathematics,*
*Shenzhen Research Institute of Big Data,*
*The Chinese University of Hong Kong, Shenzhen*

**He Zhao**                                                      *he.zhao@data61.csiro.au*
*CSIRO's Data61, Australia*

**Mingyuan Zhou**                                                      *mingyuan.zhou@mccombs.utexas.edu*
*The University of Texas at Austin, USA*

**Hongyuan Zha** [*]                                                      *zhahy@cuhk.edu.cn*
*The Chinese University of Hong Kong, Shenzhen*

**Reviewed on OpenReview:** *https://openreview.net/forum?id=QOrzmDQYou*

## Abstract

Large-scale datasets for training many real-world machine learning models pose significant computational resource challenges. One approach to mitigate this is via data condensation, which aims at learning a small dataset but still sufficiently capturing the rich information in the original one. Most of the existing approaches learn the condensed dataset and task-related model parameters (*e.g.*, classifier) in a bi-level meta-learning way. The recently proposed distribution matching (DM), however, avoids the expensive bi-level optimization but ignores task-related models. This work proposes a novel class preserving DM framework consisting of two key components. The first one is responsible for capturing the original data distribution of each class based on energy distance, which can encourage diversity in the generated synthetic data. The other is the classifier-critic constraint, which forces the learned synthetic samples to fit pre-trained task-related models, such as an off-the-shelf classifier. By designing the optimization loss in this way, we can generate more diverse and class-preserving distilled data without bi-level optimization. Extensive experiments reveal that our method can produce more effective condensed data for downstream tasks with less training cost and can also be successfully applied to de-biased dataset condensation[1].

## 1 Introduction

Deep neural networks (DNNs) have demonstrated unprecedented results in many applications and come with a cost: the training of DNNs heavily relies on the sheer amount of data, sometimes up to tens of millions of samples. Although it becomes easier than ever to construct large scale datasets with advanced data collection and labeling tools, the rapidly growing size of datasets not only posts challenges to data storage and preprocessing, but also makes it increasingly expensive to train a model on the given large-scale

---

[*]Corresponding author
[1]Code is available on `https://github.com/BIRlz/TMLR_Dataset_Condensation`

dataset. More importantly, designing new deep learning models or applying them to new tasks certainly require substantially more computations, as they involve to train multiple models on the same dataset for many times to verify the design choices, such as loss functions, architectures and hyperparameters (Cui et al., 2022; Ying et al., 2019; Lee et al., 2022). As a result, there is a strong demand for techniques that compress a large-scale dataset into a small subset of informative examples.

Numerous research endeavours have, therefore, focused on alleviating the cumbersome training process through constructing small training sets. One well-studied approach is referred to as coreset or subset selection (Agarwal et al., 2004; Sener & Savarese, 2018), which chooses important data points for training based on some heuristic criteria. However, most selection procedures incrementally and greedily select samples, which are shortsighted and do not guarantee any optimal solution for the downstream tasks. Besides, the presence of representative samples is not guaranteed (Zhao & Bilen, 2021; Zhao et al., 2021). Instead of selecting from existing data points, recently, *dataset condensation* (DC) (Wang et al., 2018; Zhao et al., 2021; Nguyen et al., 2021a;b; Zhao & Bilen, 2021; 2023; Lee et al., 2022; Cazenavette et al., 2022; Wang et al., 2022) has emerged as a competitive alternative with promising results, which aims to condense a small training set $\mathcal{S}$ from a large-scale one $\mathcal{T}$ so that models trained on $\mathcal{S}$ can generalize to test data. Once the condensed dataset is learned, one can use it for various downstream classification applications, such as implementing neural architecture search (NAS) (Dong & Yang, 2020). Compared to performing NAS with the original large-scale dataset, using the small condensed dataset for capturing the rich information in the original one does save computational resources, which is the goal of data condensation.

Along this line, a meta-learning-based strategy is proposed (Wang et al., 2018) followed by lots of data condensation methods. Among them, methods employing feature or distribution matching (DM) (Zhao & Bilen, 2023; Zhao et al., 2023) have gained prominence due to their ability to maintain competitive accuracy while requiring relatively low computational resources. Different from bi-level optimization based frameworks (Zhao et al., 2021; Zhao & Bilen, 2021; Lee et al., 2022; Cazenavette et al., 2022; Wang et al., 2022; Loo et al., 2022; 2023; Feng et al., 2023), DM-based techniques eliminate the requirement for nested optimization. In terms of DM-based methods, synthesized dataset is usually optimized by minimizing the distributional distance between real dataset and synthetic synthetic dataset. Therefore, a key challenge in DM is how to design an effective metric for measuring the distributional distance. For example, Zhao & Bilen (2023); Zhao et al. (2023); Zhang et al. (2024) adopt Maximum Mean Discrepancy (MMD) (Gretton et al., 2012) as the metric; Yin et al. (2023) propose a matching metric based on batch normalization statistics; and Wang et al. (2025) introduce Neural Characteristic Function Discrepancy.

In this paper, we stress that a condensed dataset should preserve the class information of the original dataset and its samples should be diverse enough to represent the original dataset, so that a model trained on the condensed dataset is generalizable to test data. Taking the DM (Zhao & Bilen, 2023) as the example, it generates a condensed dataset independently for each class, without guaranteeing that the synthetic samples actually belong to the class. In other words, the synthetic samples might ignore some discriminative features of their corresponding class, which will harm their availability in various downstream classification tasks. Moreover, the MMD distance used in DM ignores the diversity of the generated samples. As stated by Sun et al. (2024), a high-quality should cover a wide range of samples and labels, which is essential for robust learning and generalization. That is to say, diversity is a key point when we learning the synthetic set since we expect each condensed sample plays a different role for reducing redundancy. To this end, we propose a new distribution matching framework to satisfy above two factors simultaneously. To enhance the class-preserving property of the condensed datasets, inspired by the idea of plug-and-play generative models (Nguyen et al., 2015; 2017), we propose to pre-train an off-the-shelf model to capture the class information in the training set and then use it to derive a classifier-critic regularization on the synthetic dataset. Designed in this way, the learning of classification model and synthetic set is disentangled, which avoids the bi-level optimization. However, moving beyond DM, we can preserve the class information of the original dataset by learning the condensed samples from the view of the classifier-critic constraint. To encourage the synthetic samples to be more diverse, we introduce to minimize the energy distance (Székely et al., 2007; Rizzo & Székely, 2016) between the synthetic samples and the real ones. In addition to making synthetic samples close to the real samples, energy distance enjoys an intrinsic mechanism to encourage the synthetic samples to be different from each other. Due to its flexibility, our proposed loss can be easily

combined with most of existing distribution matching methods, which can further enhance the quality of synthetic datasets. In the comprehensive experiments, ours achieves the competing performance on standard dataset condensation tasks compared with other related methods. In addition, our proposed classifier-critic constraint enables our method to generate synthetic samples that preserve other necessary information from the original dataset, such as in the case of de-biased dataset condensation.

We summarize our contributions as follows: (1) We introduce energy distance to as the DM loss, for measuring the distributional distance between real and synthetic samples. (2) We propose a class preserving distribution matching approach for optimizing the condensed dataset, where we introduce a classifier-critic regularization and combine it with DM loss. (3) As a plug-and-play matching loss, we instantiate how to combine ours with classical or recent popular data condensation methods. (4) We conduct extensive experiments under diverse scenarios and consider de-biased dataset condensation due to the flexibility of classifier-critic constraint.

## 2 Related Work

### 2.1 Data Condensation Methods

**Bi-level Optimization Methods.** Bi-level dataset distillation methods formulate dataset condensation as a nested optimization problem, where the task model is trained in the inner loop and the synthetic data is optimized in the outer loop. Among them, a representative class of such methods is *Meta-Model Matching*, which directly optimizes the transferability or generalization performance of models trained on synthetic data. Typical approaches include Dataset Distillation (DD) (Wang et al., 2018), Kernel Inducing Points (KIP) (Nguyen et al., 2021b), FRePo (Zhou et al., 2022), and LinBa (Deng & Russakovsky, 2022). *Gradient Matching* aims to optimize the synthetic dataset by matching gradients between real and synthetic data, such as Dataset Condensation (DC) (Zhao et al., 2021), Differentiable Siamese Augmentation (DSA) (Zhao & Bilen, 2021), DC with contrastive signals (DCC) (Lee et al., 2022) and Information-intensive Dataset Condensation (IDC) (Kim et al., 2022). *Trajectory Matching* matches the training trajectories of models trained on original and synthetic data in multiple steps, such as MTT (Cazenavette et al., 2022) and TESLA (Cui et al., 2023) , FTD (Du et al., 2023), ATT Liu et al. (2024), etc. To improve efficiency, (Loo et al., 2022; 2023) introduce random feature approximations and convexified implicit gradients; (Feng et al., 2023) simplifies unrolled optimization to reduce computational cost while retaining effectiveness.

**Distribution Matching Methods.** Different from bi-level optimization methods, DM methods aim to align the statistical or structural distributions of real and synthetic data, without involving the nested model optimization. Some methods, including DD, Zhao & Bilen (2023), M3D (Zhang et al., 2024)and IDM (Zhao et al., 2023), aim to minimize the maximum mean discrepancy (MMD) between synthetic and real datasets. Some methods also align feature distributions without explicitly optimizing statistical distances, such as CAFE (Wang et al., 2022), Datadam (Sajedi et al., 2023), IID (Deng et al., 2024). Yin et al. (2023) propose to match batch normalization statistics and relabeling synthetic data using pretrained classifiers. Wang et al. (2025) reformulate distribution matching as a minmax optimization problem and propose neural characteristic function discrepancy (NCFD) as the distance measure between datapoints.

Ours falls into the group of distribution matching. In contrast to existing DM methods, which either minimize MMD (e.g., in DD, M3D, IDM) or align feature distributions implicitly (e.g., CAFE, Datadam, IID), our method introduces a flexible and fundamental loss based on Energy Distance (ED). ED can simultaneously encourage real-synthetic alignment and intra-synthetic diversity. To further enhance class separability and optimization stability, we incorporate a lightweight classifier loss as a regularization term. As an explicit loss, our proposed method is flexible, compatible with learnable feature mappings like NCFD, and can also be easily integrated as an auxiliary loss into broader optimization frameworks.

**Diverse Dataset Condensation Methods.** Recent works have highlighted the importance of diversity in dataset condensation to avoid mode collapse and improve generalization. MMDiff (Gu et al., 2024) formulates distillation as a minimax optimization over a diffusion-inspired potential, encouraging synthetic samples to spread across the data manifold efficiently. DSDM (Li et al., 2024) introduces class-wise semantic distribution matching to enhance diversity. It aligns real and synthetic feature distributions while promoting intra-class variation in the semantic space. RDED (Sun et al., 2024) balances sample diversity and realism via

a dual-objective loss, combining inter-sample repulsion with perceptual realism constraints from pretrained discriminators. Compared to these, our method promotes diversity via the repulsive term in the energy distance, while maintaining semantic consistency through a classifier-guided constraint and the matching part in energy distance. Importantly, our design is modular and can be flexibly combined with various objectives, without relying on generators or bi-level training.

## 2.2 Selection-based Methods

The classic technique to compress the training set is coreset or subset selection (Agarwal et al., 2004; Chen et al., 2010; Wei et al., 2015). In addition to one naive method, which randomly picks data from the original dataset, most of these methods incrementally select important data points based on some heuristic selection criteria. For example, (Sener & Savarese, 2018) select data points such that the largest distance between a data point and its nearest center is minimized. (Aljundi et al., 2019) use the parameter gradient as the feature to maximize the diversity of samples in the replay buffer. However, these heuristic selection criteria cannot ensure that the selected subset is optimal for the downstream task, especially for training DNNs. Besides, the information in the dataset is usually uniformly distributed over all samples, thus finding such an informative coreset may not always be possible (Zhao & Bilen, 2021; Zhao et al., 2021).

## 2.3 Plug and Play Generative models

Our work is also closely related to generative models such as variational auto-encoder (Kingma & Welling, 2014) and generative adversarial networks (GANs) (Goodfellow et al., 2014). The difference is that image generation aims to generate real-looking and high-fidelity images that can fool human eyes. As explored by (Zhao et al., 2021), the images produced by GANs have similar performances to those randomly selected real images and are usually weaker than the dataset condensation methods. Another related work is plug-and-play generative networks (PPGN) (Nguyen et al., 2017; Graikos et al., 2022), which composes of 1) a generator for drawing a wide range of image types, and 2) a replaceable "condition" network that tells the generator what to draw, similar to our classifier-critic constraint. Despite of the recent community interest in PPGN, the possibility of the constraint in dataset condensation has not been explored. Instead of producing "real-looking" samples under some constraints, we aim to generate a condensed informative training set for the target task.

# 3 Preliminary

## 3.1 Energy distance

Energy distance is a statistical distance between probability distributions (Rizzo and Szekely ,2016). For two independent random vectors $\mathbf{X}$ and $\mathbf{Y}$ in $R^d$, the energy distance between them is defined as

$$\varepsilon(\mathbf{X}, \mathbf{Y}) = 2E\|\mathbf{X} - \mathbf{Y}\| - E\|\mathbf{X} - \mathbf{X}'\| - E\|\mathbf{Y} - \mathbf{Y}'\|$$

where $E$ means expectation operator, $\|\cdot\|$ is the Euclidean norm, $E\|\mathbf{X}\| < \infty, E\|\mathbf{Y}\| < \infty, \mathbf{X}'$ and $\mathbf{Y}'$ denote the independent and identically distributed (iid) copy of $\mathbf{X}$ and $\mathbf{Y}$, respectively.

A significant advantage of the energy distance is that $\varepsilon(\mathbf{X}, \mathbf{Y}) = 0$ if and only if $\mathbf{X}$ and $\mathbf{Y}$ are identically distributed. Thus, the energy distance can be used for testing of equal distributions or multivariate goodness-of-fit measure. Another advantage of the energy distance is that it is distribution free. That is to say, the estimated value of energy distance does not depend on the distribution form of random vectors, although it can be represented as the form of characteristic function. Therefore, the energy distance can be estimated with the following surprisingly simple form. Denoting $\mathbf{x_1}, \ldots, \mathbf{x_{n_1}}$ as the samples of $\mathbf{X}$ and $\mathbf{y_1}, \ldots, \mathbf{y_{n_2}}$ as the samples of $\mathbf{Y}$, we can estimate the energy distance as follows

$$\varepsilon_{n_1,n_2}(\mathbf{X}, \mathbf{Y}) = 2\frac{1}{n_1 n_2}\sum_{i=1}^{n_1}\sum_{j=1}^{n_2}\|\mathbf{x}_i - \mathbf{y}_j\| - \frac{1}{n_1^2}\sum_{i=1}^{n_1}\sum_{j=1}^{n_2}\|\mathbf{x}_i - \mathbf{x}_j\| - \frac{1}{n_2^2}\sum_{i=1}^{n_1}\sum_{j=1}^{n_2}\|\mathbf{y}_i - \mathbf{y}_j\| \tag{1}$$

### 3.2 Previous Works: Distribution Matching Methods for Dataset Condensation

Denote a large-scale dataset as $\mathcal{T} = \{\boldsymbol{x}_i, \boldsymbol{y}_i\}|_{i=1}^{|\mathcal{T}|}$ with $|\mathcal{T}|$ image and label pairs and $C$ classes, and denote the small (synthetic) dataset as $\mathcal{S} = \{\boldsymbol{s}_j, \boldsymbol{y}_j\}|_{j=1}^{|\mathcal{S}|}$. We can further represent the real set and synthetic set in class $c$ as $\mathcal{T}_c$ and $\mathcal{S}_c$, respectively. DM (Zhao & Bilen, 2023) learn $\mathcal{S}_c$ by minimizing the empirical estimate of MMD between the real and synthetic samples of the class $c$:

$$E_{\theta \sim P_\theta} \left\| \frac{1}{|\mathcal{T}_c|} \sum_{i=1}^{|\mathcal{T}_c|} f_\theta\left(\boldsymbol{x}_i\right) - \frac{1}{|\mathcal{S}_c|} \sum_{j=1}^{|\mathcal{S}_c|} f_\theta\left(\boldsymbol{s}_j\right) \right\|^2, \tag{2}$$

where $f_\theta$ is the embedding function parameterized with $\theta$ sampled from $P_\theta$, $|\mathcal{T}_c|$ and $|\mathcal{S}_c|$ is the number of samples of $\mathcal{T}_c$ and $\mathcal{S}_c$. DM obtains the whole synthetic dataset $\mathcal{S}$ by summarizing $\mathcal{S}_c$ in each class. Different from others (Zhao & Bilen, 2021; Zhao et al., 2021; Wang et al., 2022), DM avoids the expensive bi-level optimization and second-order derivative with promising performance. However, DM ignores whether the synthetic set $\mathcal{S}_c$ can actually be helpful to the classification of the class $c$ when optimizing them. Moreover, DM ignores the diversity of distilled samples when using MMD and produces undesired performance with a slightly larger compression ratio.

## 4 Method

### 4.1 Proposed Method: Class-preserving Distribution Matching

In this work, we explore a novel efficient dataset condensation approach with the following intuition: a condensed synthetic set is desired if the generated samples can not only match the distribution of original training set but also can be confidently classified its corresponding labels by a well-trained classifier. Therefore, we introduce a distribution matching loss and a classifier-critic constraint to learn the condensed dataset $\mathcal{S}$, described below in detail.

### 4.2 Classifier-critic Constraint

Existing dataset condensation methods put their attention on the standard dataset condensation task, where they assume the original large-scale training set is high-quality without subpopulation shift or other problems. However, datasets in real world are usually not perfect. Below, we first introduce the classifier-critic constraint for standard (commonly-used) dataset condensation and then explain how to design the corresponding classifier-critic constraint for de-biased dataset condensation.

**Standard dataset condensation.** Specifically, we can assume $g_\phi$ is a classifier pre-trained on the training set $\mathcal{T}$, $e.g.$, based on the Cross-Entropy (CE) loss. Then we can use the given classifier $g_\phi$ to design a classifier-critic regularization with CE loss when optimizing each synthetic samples $\boldsymbol{s}$ with label $\boldsymbol{y}$, formulated as:

$$\ell = -\log \Pr(\boldsymbol{y} \mid \boldsymbol{s}; g_\phi) \tag{3}$$

where $g_\phi$ is frozen during the learning of $\mathcal{S}$ since it already has the acceptable classification ability. In addition to using a classifier pre-trained on $\mathcal{T}$, we can also adopt existing available classifers, such as the powerful Contrastive Language¨CImage Pre-training (CLIP) model (Radford et al., 2021), which efficiently learns visual concepts from natural language supervision based a sufficiently large dataset rather than $\mathcal{T}$. This constraint forces the to-be-learned synthetic samples $\boldsymbol{s}$ being classified into its corresponding label $\boldsymbol{y}$ using the pre-trained classifier $g_\phi$. That is to say, we consider distilling the discriminative information beneficial for the classification task into the synthetic set $\mathcal{S}$.

**De-biased dataset condensation.** Subpopulation shift widely exists in many real-world applications, where subpopulations are seen but underrepresented in the training data (Han et al., 2022; Yao et al., 2022), such as fairness of machine learning and class imbalance (Hashimoto et al., 2018; Japkowicz, 2000). Then models may perform poorly when they falsely rely on the spurious correlation between the particular subpopulation and the label. Thus, a practical research is condensing a training dataset with biased demographic

subpopulations into a small de-biased synthetic dataset. Here $A$ denotes the number of spurious attributes like gender and ethnicity, where we can represent the attribute vector of synthetic sample $\boldsymbol{s}$ as $\boldsymbol{a}$. Now we can pre-train two classifiers on original dataset $\mathcal{T}$, where one is the label classifier $g_{\phi_1}$ for predicting class while other is spurious attribute classifier $g_{\phi_2}$. To enforce the presence of label and absence of the spurious attribute, we can define the classifier-critic as follows:

$$\ell = -\log \Pr(\boldsymbol{y} \mid \boldsymbol{s}; g_{\phi_1}) + \log \Pr(\boldsymbol{a} \mid \boldsymbol{s}; g_{\phi_2}). \tag{4}$$

By minimizing the CE loss about label and maximizing the CE loss about the spurious attribute, we can encourage the synthetic image to be classified as its label and not be recognized as its spurious attribute. It reduces the spurious correlations of label on subpopulation group in condensed samples.

### 4.3 Energy Distance between $\mathcal{S}_c$ and $\mathcal{T}_c$

Enforcing the matching between $\mathcal{S}_c$ and $\mathcal{T}_c$ is also important, otherwise we might encounter "spurious" examples that have high classification performance only under pre-trained classifier but ignore the structure in data, which may cause poor generalization performance when using $\mathcal{S}$ to train downstream classification tasks. Different from generative models that aim to generates real-looking images, dataset condensation aims to accurately match the distribution of the real training data with limited synthetic data. Despite DM learning $\mathcal{S}$ with MMD and proving its effectiveness, it may ignore the diversity in the synthesized samples, which plays a key role in reducing the redundancy of condensed samples. To this end, we propose to learn $\mathcal{S}$ by minimizing the energy distance between the real and synthetic data distributions for each class. We first introduce the definition of energy distance (Rizzo & Székely, 2016). Denoting $\mathbf{X}$ and $\mathbf{Y}$ as two independent random vectors in $R^d$, their energy distance is defined as:

$$\varepsilon(\mathbf{X}, \mathbf{Y}) = 2E\|\mathbf{X} - \mathbf{Y}\| - E\|\mathbf{X} - \mathbf{X}'\| - E\|\mathbf{Y} - \mathbf{Y}'\|, \tag{5}$$

where $\|\cdot\|$ indicates the Euclidean norm, $E\|\mathbf{X}\| < \infty, E\|\mathbf{Y}\| < \infty$, $\mathbf{X}'$ and $\mathbf{Y}'$ denote the independent and identically distributed (iid) copy of $\mathcal{X}$ and $\mathcal{Y}$, respectively. A significant advantage of the energy distance is that $\varepsilon(\mathbf{X}, \mathbf{Y}) = 0$ if and only if $\mathbf{X}$ and $\mathbf{Y}$ are identically distributed.

In this work, taking the class $c$ as the example, we have samples $\boldsymbol{x}$ and $\boldsymbol{s}$ from $\mathcal{T}_c$ and $\mathcal{S}_c$, respectively. Now the energy distance between $\mathcal{T}_c$ and $\mathcal{S}_c$ can be estimated as follows:

$$\varepsilon(\mathcal{T}_c, \mathcal{S}_c) = \frac{2}{|\mathcal{T}_c||\mathcal{S}_c|} \sum_{i=1}^{|\mathcal{T}_c|} \sum_{j=1}^{|\mathcal{S}_c|} d(\boldsymbol{x}_i, \boldsymbol{s}_j) - \frac{1}{|\mathcal{S}_c|^2} \sum_{i,j=1}^{|\mathcal{S}_c|} d(\boldsymbol{s}_i, \boldsymbol{s}_j) - \frac{1}{|\mathcal{T}_c|^2} \sum_{i,j=1}^{|\mathcal{T}_c|} d(\boldsymbol{x}_i, \boldsymbol{x}_j), \tag{6}$$

where we compute the distance function in embedding spaces following DM (Zhao & Bilen, 2023), *i.e.* $d(\boldsymbol{x}_i, \boldsymbol{s}_j) = \|f_\theta(\boldsymbol{x}_i) - f_\theta(\boldsymbol{s}_j)\|$ with $f_\theta$ denoting the feature extractor sampled from $P_\theta$. The main difference between Eq. (2) in DM and our adopted energy distance is the presence of a *repulsive* term between generated data, $d(\boldsymbol{s}_i, \boldsymbol{s}_j)$, which encourages the condensed samples to capture the full distribution and improve the representativeness.

Discussion about why adding the energy distance can help improve diversity: Recall that our adopted energy distance in Eq (5) is composed of $\sum_{i,j=1} d(x_i, s_j)$, $\sum_{i,j=1} d(s_i, s_j)$ and $\sum_{i,j=1} d(x_i, x_j)$, where the third term can be ignored for only computing the distance between real samples, i.e., $x_i$ and $x_j$. The first term $\sum_{i,j=1} d(x_i, s_j)$ aims to minimize the distance between real and synthetic samples, which is the attractive term similar to the empirical estimate of MMD in Eq (1). The second term $\sum_{i,j=1} d(s_i, s_j)$ of ED in Eq (5) aims to maximize the distance between the condensed data points, playing the role of a repulsive term. Therefore, the main difference between the adopted ED in Eq (5) and the empirical estimate of MMD in Eq (1) is the repulsive term. And we had conducted the ablation study to compare the ED and MMD (i.e., DM) in Table 2 in experiments, where ED produces superior performance than that of DM. Especially, ED outperforms DM by large margins with larger IPCs. It is reasonable since ED not only minimizes the distance between synthetic data and real dataset, but also maximizes the distance between synthetic samples. Therefore, we stress that ED enjoys an intrinsic mechanism to encourage the synthetic samples to be different from each other, improving the diversity of synthetic samples.

---

**Algorithm 1** Workflow of our method.

---

**Require:** Training set $\mathcal{T}$, synthetic set $\mathcal{S}$ randomly initialized from $\mathcal{T}$ with corresponding labels, off-the-shelf classifier $g_\phi$, feature extractor $f_\theta$, number of classes C, max-iter and learning rate $\eta$.

**for** $t$ in maxiter **do**

    Randomly sample $\theta$ from $P_\theta$ and compute $\lambda$;

    Randomly sample a minibatch $B_c^{\mathcal{T}} \sim \mathcal{T}_c$ and $\mathcal{S}_c \sim \mathcal{S}$ for each class c in $C$;

    Compute $\mathcal{L} = \sum_c^C \left( \varepsilon(B_c^{\mathcal{T}}(\theta), \mathcal{S}_c(\theta)) - \lambda E_{\boldsymbol{s} \sim \mathcal{S}_c}[\Pr(\boldsymbol{y} \mid \boldsymbol{s}; g_\phi)] \right)$

    Update $\mathcal{S} \leftarrow \mathcal{S} - \eta \nabla_{\mathcal{S}} \mathcal{L}$;

**end for**

**Output:** a synthetic dataset $\mathcal{S}$

---

### 4.4 Training Loss

Recall that we aim to learn the class preserving and diverse condensed samples, which can be solved by introducing a classifier-critic constraint and the energy distance between $\mathcal{T}_c$ and $\mathcal{S}_c$. Therefore, to learn the synthetic set $\mathcal{S}_c$ of class $c$ for standard (or de-biased) dataset condensation, we can jointly minimize the energy distance in (6) and the CE loss in (3) (or in (4)). Therefore, the optimization loss can be formulated as follows:

$$\min_{\mathcal{S}_c} \mathcal{L} = E_{\theta \sim P_\theta}[\varepsilon(\mathcal{T}_c, \mathcal{S}_c)] - \lambda E_{\boldsymbol{s} \sim \mathcal{S}_c}[\log \Pr(\boldsymbol{y} \mid \boldsymbol{s}; g_\phi)], \tag{7}$$

where the trade-off hyper-parameter $\lambda = \frac{\text{maxiter}-t}{\text{maxiter}}$ with $t$ denoting the $t$-th iteration and maxiter denoting the maximum number of iterations. We summarize the training process of standard dataset condensation in Algorithm 1, which targets at image classification problems. Following Zhao & Bilen (2023; 2021), we initialize the synthetic images using randomly sampled real images with corresponding labels before training. At each training iteration, we can randomly sample embedding function $\theta$ from $P_\theta$ and a mini-batch of real samples for each class. Our method enforces the synthetic data to assimilate the task-related information by a very flexible classifier-critic constraint parameterized with already-learned $\phi$, avoiding a bi-level optimization and second-order derivative. It is different from DM which ignores classifier-related model when learning the synthesized dataset, *i.e.* without considering the classification performance of the synthetic samples; and also different from other related work (Zhao et al., 2021; Wang et al., 2022; Zhao & Bilen, 2021) that usually couples the learning of the synthesized set with the learning of task-related model at each training iteration. Besides, it improves the diversity of the condensed samples by introducing the energy distance.

## 5 Combination between Ours and Baselines

As discussed above, our proposed loss is flexible for being an explicit loss. Therefore, it can be easily combined with other methods. Below, we consider two specific examples.

### 5.1 Combining Characteristic Function Distance with Energy Distance in Ours

**Characteristic Function Distance (CFD).** To better compare probability distributions, we consider the *characteristic function distance* (CFD) proposed by Wang et al. (2025) as an alternative to traditional pairwise distances. Given a random variable $x \in R^d$, its characteristic function is defined as the Fourier transform of its probability distribution:

$$\Phi_x(t) = E_x\left[e^{it^\top x}\right], \quad t \in R^d \tag{8}$$

This representation uniquely determines the distribution and has been widely used in distribution testing and generative modeling. Given two samples $x$ and $\tilde{x}$, the characteristic function distance is defined as:

$$\mathcal{C}_{\mathcal{T}}(x, \tilde{x}) = \int_t \sqrt{\left(\Phi_x(t) - \Phi_{\tilde{x}}(t)\right)\left(\overline{\Phi_x(t)} - \overline{\Phi_{\tilde{x}}(t)}\right)} \, dF_{\mathcal{T}}(t) \tag{9}$$

In practice, this integral can be estimated by Monte Carlo sampling over $t$, and each input is passed through a learnable feature extractor $f_\theta$, enabling learnable and expressive distance modeling.

**CFD-enhanced Energy Distance.** We propose to incorporate CFD into the empirical energy distance structure. Recall the empirical energy distance in Eq. (6) , we replace the Euclidean distance $d(\cdot, \cdot)$ with our proposed CFD and can replace the first term in Eq.(7) with following equation:

$$
\begin{aligned}
\mathcal{L}_{\text{CFD-ED}}(T_c, S_c) = 2 \cdot E_{x \sim T_c, s \sim S_c} \mathcal{C}_\mathcal{T}(x, s; f_\theta, \psi) \\
- E_{s, s' \sim S_c} \mathcal{C}_\mathcal{T}(s, s'; f_\theta, \psi) - E_{x, x' \sim T_c} \mathcal{C}_\mathcal{T}(x, x'; f_\theta, \psi)
\end{aligned}
\tag{10}
$$

where $\mathcal{C}_\mathcal{T}(x, \tilde{x}; f_\theta, \psi)$ is the characteristic function distance computed in the feature space:

$$
\mathcal{C}_\mathcal{T}(x, \tilde{x}; f_\theta, \psi) = \int_t \sqrt{\left( \Phi_{f_\theta(x)}(t) - \Phi_{f_\theta(\tilde{x})}(t) \right) \left( \overline{\Phi_{f_\theta(x)}(t)} - \overline{\Phi_{f_\theta(\tilde{x})}(t)} \right)} \, dF_\mathcal{T}(t; \psi),
\tag{11}
$$

where projection parameter $\psi$ governs the distribution or transformation of frequency inputs $t$, which are treated as learnable or fixed random variables; please see more details in Wang et al. (2025). The formulation in Eq.(10) inherits the structural benefits of energy distance and characteristic function distance.

## 5.2 Ours Serving as a Regularization Term

The M3D method (Zhang et al., 2024) performs dataset condensation by minimizing the discrepancy between real and synthetic data distributions in a reproducing kernel Hilbert space (RKHS). It introduces an efficient Maximum Mean Discrepancy (MMD) loss to match higher-order statistics without relying on a classifier. Given a real data batch $T_c$ and a synthetic data batch $S_c$ for class $c$, the M3D loss is defined as:

$$
\mathcal{L}_{\text{M3D}}(T_c, S_c) = \frac{1}{|T_c|^2} \sum_{x, x' \in T_c} k(f_\theta(x), f_\theta(x')) + \frac{1}{|S_c|^2} \sum_{s, s' \in S_c} k(f_\theta(s), f_\theta(s')) - \frac{2}{|T_c||S_c|} \sum_{x \in T_c, s \in S_c} k(f_\theta(x), f_\theta(s))
\tag{12}
$$

where $k(\cdot, \cdot)$ is a universal kernel function (e.g., Gaussian RBF kernel $k(x, x') = exp(-\lambda||x - x'||^2)$); please find more details from Zhang et al. (2024). While effective for global distribution matching, M3D lacks explicit mechanisms for semantic preservation and intra-class diversity. To address these, our proposed loss in Eq. (7) can serve as the regularization term of M3D. The resulting overall objective is:

$$
\mathcal{L}_{\text{total}} = \mathcal{L}_{\text{M3D}} + \alpha \mathcal{L},
\tag{13}
$$

where $\alpha$ is trade-off hyperparameter and we set as $\alpha = 1$. This formulation unifies model-agnostic distribution matching with semantic consistency and diversity control. Besides, ours can also be combined with SRe$^2$L that minimizing the distance of batch normalization statistics between real and synthetic datasets (Yin et al., 2023). Similar to M3D+OURS, SRe$^2$L+OURS is implemented by introducing our loss as the regularization loss of SRe$^2$L when optimizing the condensed images.

# 6 Experiments

## 6.1 Datasets and Implementation Details

**Datasets.** For standard dataset condensation, we consider two widely adopted image datasets, including CIFAR10 and CIFAR100. **CIFAR10** (Krizhevsky et al., 2009) and **CIFAR100** (Krizhevsky et al., 2009) consists of tiny colored natural images from 10 and 100 categories, respectively. Each dataset has 50K training images and 10K test images with the size of $32 \times 32$. Besides, following Yin et al. (2023), we also evaluate our proposed method on large-scale TinyImageNet. **TinyImageNet** (Le & Yang, 2015) incorporates 200 classes derived from ImageNet1K, with each class comprising 500 images processing a resolution $64 \times 64$.

**Implementation Details.** Following previous work (Zhao & Bilen, 2023; Wang et al., 2025), we adopt the commonly-used DSA augmentation (Zhao & Bilen, 2021) and learn 1/10/50 image(s) per class as (IPC) synthetic sets for CIFAR-10/100 using the ConvNet architecture. The ConvNet includes three repeated convolutional blocks, and each block involves a 128-kernel convolution layer, instance normalization layer (Ulyanov et al., 2016), ReLU activation function (Nair & Hinton, 2010) and average pooling. For large-scale

Table 1: Comparison with baseline models on CIFAR-10/100. We cite the results from Wang et al. (2025).

| Category | Method | CIFAR-10 | | | CIFAR-100 | | |
|---|---|---|---|---|---|---|---|
| | | 1 (0.02%) | 10 (0.2%) | 50 (1%) | 1 (0.2%) | 10 (2%) | 50 (10%) |
| Traditional | Random | $14.4_{\pm2.0}$ | $26.0_{\pm1.2}$ | $43.4_{\pm1.0}$ | $4.2_{\pm0.3}$ | $14.6_{\pm0.5}$ | $30.0_{\pm0.4}$ |
| | Herding | $21.5_{\pm1.2}$ | $31.6_{\pm0.7}$ | $40.4_{\pm0.6}$ | $8.4_{\pm0.3}$ | $17.3_{\pm0.3}$ | $33.7_{\pm0.5}$ |
| | Forgetting | $13.5_{\pm1.2}$ | $23.3_{\pm1.0}$ | $23.3_{\pm1.1}$ | $4.5_{\pm0.2}$ | $15.1_{\pm0.3}$ | $30.5_{\pm0.3}$ |
| Diverse | DSDM | $45.0_{\pm0.4}$ | $66.5_{\pm0.3}$ | $75.8_{\pm0.3}$ | $19.5_{\pm0.2}$ | $46.2_{\pm0.3}$ | $54.1_{\pm0.2}$ |
| Bi-level | DC | $28.3_{\pm0.5}$ | $44.9_{\pm0.5}$ | $53.9_{\pm0.5}$ | $12.8_{\pm0.3}$ | $25.2_{\pm0.3}$ | |
| | DSA | $28.8_{\pm0.7}$ | $52.1_{\pm0.5}$ | $60.6_{\pm0.5}$ | $13.9_{\pm0.3}$ | $32.3_{\pm0.4}$ | $42.8_{\pm0.4}$ |
| | DCC | $32.9_{\pm0.8}$ | $49.4_{\pm0.5}$ | $61.6_{\pm0.4}$ | $13.3_{\pm0.3}$ | $30.6_{\pm0.4}$ | $40.0_{\pm0.3}$ |
| | MTT | $46.3_{\pm0.8}$ | $65.3_{\pm0.7}$ | $71.6_{\pm0.2}$ | $24.3_{\pm0.3}$ | $40.1_{\pm0.4}$ | $47.7_{\pm0.2}$ |
| | TESLA | $48.5_{\pm0.8}$ | $66.4_{\pm0.8}$ | $72.6_{\pm0.4}$ | $24.8_{\pm0.4}$ | $41.7_{\pm0.3}$ | - |
| | FTD | $46.8_{\pm0.3}$ | $66.6_{\pm0.3}$ | $73.8_{\pm0.2}$ | $25.2_{\pm0.2}$ | $43.4_{\pm0.3}$ | $48.5_{\pm0.3}$ |
| | ATT | $48.3_{\pm1.0}$ | $67.7_{\pm0.6}$ | $74.5_{\pm0.2}$ | $26.1_{\pm0.4}$ | $44.0_{\pm0.5}$ | $51.2_{\pm0.3}$ |
| | FrePo | $46.8_{\pm0.7}$ | $65.5_{\pm0.4}$ | $71.7_{\pm0.2}$ | $28.7_{\pm0.2}$ | $42.5_{\pm0.2}$ | $44.3_{\pm0.2}$ |
| Distribution matching | CAFE | $30.3_{\pm1.1}$ | $46.3_{\pm0.6}$ | $55.5_{\pm0.6}$ | $12.9_{\pm0.2}$ | $27.8_{\pm0.3}$ | $37.9_{\pm0.3}$ |
| | IDM | $45.6_{\pm0.7}$ | $58.6_{\pm0.1}$ | $67.5_{\pm0.2}$ | $20.1_{\pm0.3}$ | $45.1_{\pm0.1}$ | $50.0_{\pm0.2}$ |
| | DM | $26.0_{\pm0.8}$ | $48.9_{\pm0.6}$ | $63.0_{\pm0.4}$ | $11.4_{\pm0.3}$ | $29.7_{\pm0.3}$ | $43.6_{\pm0.4}$ |
| | **OURS** | $29.3_{\pm0.3}$ | $53.4_{\pm0.2}$ | $65.9_{\pm0.3}$ | $12.8_{\pm0.2}$ | $33.4_{\pm0.2}$ | $45.2_{\pm0.2}$ |
| | M3D | $45.3_{\pm0.3}$ | $63.5_{\pm0.2}$ | $69.9_{\pm0.2}$ | $26.2_{\pm0.3}$ | $42.4_{\pm0.2}$ | $50.9_{\pm0.7}$ |
| | **M3D+OURS** | $45.9_{\pm0.2}$ | $63.2_{\pm0.1}$ | $71.0_{\pm0.3}$ | $27.6_{\pm0.2}$ | $43.2_{\pm0.4}$ | $52.5_{\pm0.5}$ |
| | NCFM (Reproduced) | $47.0_{\pm0.2}$ | $\mathbf{69.9}_{\pm0.6}$ | $77.80_{\pm0.4}$ | $30.0_{\pm0.2}$ | $48.5_{\pm0.1}$ | $54.2_{\pm0.1}$ |
| | **NCFM + OURS** | $\mathbf{47.2}_{\pm0.5}$ | $69.6_{\pm0.4}$ | $\mathbf{77.9}_{\pm0.6}$ | $\mathbf{30.3}_{\pm0.3}$ | $\mathbf{48.7}_{\pm0.1}$ | $\mathbf{54.6}_{\pm0.2}$ |
| | Whole Dataset | | $84.8_{\pm0.1}$ | | | $56.2_{\pm0.3}$ | |

datasets, we employ ResNet18 as the backbone of TinyImageNet Yin et al. (2023). To evaluate the synthesis-based methods, we learn 5 synthetic datasets and train 20 randomly initialized classifiers on each synthetic dataset; then report the mean and standard deviation of 100 test accuracies. We set maxiter $= 40K$ and learn model parameters $\phi$ using the original training set, which adopts the same architecture and setting with the downstream classifier in the standard dataset condensation. Besides, we also utilize CLIP[2] to represent off-the-shelf model. We defer the learning rate, mini-batch size and training details for $\phi$ to Appendix A.

**Baselines.** We compare our method to (1) coreset selection methods, including Random and K-Center (Sener & Savarese, 2018). (2) Bi-level optimization methods, including DC (Zhao et al., 2021), DSA (Zhao & Bilen, 2021), DCC (Lee et al., 2022), MTT (Cazenavette et al., 2022), TESLA (Cui et al., 2023), FTD (Du et al., 2023), ATT Liu et al. (2024). (3) Distribution matching methods, including DM (Zhao & Bilen, 2023), CAFE (Wang et al., 2022), IDM (Zhao et al., 2023), M3D (Zhang et al., 2024),and SRe$^2$L (Yin et al., 2023), NCFD (Wang et al., 2025). (4) Diverse condensation methods, including DSDM (Li et al., 2024). Following Cui et al. (2022), DSA augmentation is enabled during evaluation for all base methods to make a fair comparison. For the implementation details of the baselines, please refer to dataset condensation Benchmark (Cui et al., 2022) for more details. Due to the flexibility of ours, we can also combine it with other methods, where we use their official code. Here, we consider adding our proposed loss with the matching loss in M3D and SRe$^2$L, denoted as OURS+M3D and OURS+SRe$^2$L, respectively, where we set the coefficient for two losses as 1 without exhausted adjustment. Besides, we can also use characteristic function to compute the point-wise distance in energy distance, denoted as OURS+NCFM.

## 6.2 Experimental Results on Standard Dataset Condensation

**Results on CIFAR-10/100.** We adopt a common evaluation method for dataset condensation methods, *i.e.* measuring the test accuracy of the neural networks trained on the condensed data. Here we report the performance comparison with different IPCs in Table 1 on CIFAR-10/100. While DSA and CAFE can condense more data-efficient samples when IPC=1, our method outperforms them at IPC=10/50. A

---
[2]https://github.com/openai/CLIP

similar phenomenon has been observed by DM. The possible reason is that DSA and CAFE on bigger synthetic data use limited iterations in the inner-loop model optimization to ensure scalability, making them less accurate. Despite both DM and ours adopting the distribution matching and avoiding the bi-level optimization, ours significantly outperforms DM on various settings. It is not surprising since we introduce the classifier-critic regularization and energy distance, whose effectiveness will be further explored in ablation studies. Considering our proposed method is flexible, we can also combine ours with state-of-the-art (SOTA) methods to optimize the synthetic samples. For example, we can take M3D and NCFM as two examples and explore where ours can be combined with them. We find that combining ours can usually obtain performance gains in most settings. The possible reason for the improvement is that ours loss enforces the synthetic dataset to match the real large-scale dataset in the embedding space with energy distance and classifier-critic constraint, which might be complementary to the representation matching in M3D and the characteristic function in NCFM. It demonstrates the desired generalization and flexibility of ours.

Table 2: Results of our method on Tiny-ImageNet.

| Dataset | IPC | MTT (ConvNet4) | DM(ConvNet4) | Ours(ConvNet4) | SRe$^2$L(ResNet18) | SRe$^2$L + OURS(ResNet18) |
|---|---|---|---|---|---|---|
| Tiny-ImageNet | 50 | 28.0± 0.3 | 22.7± 0.3 | 28.1± 0.2 | 41.1 ± 0.4 | **44.28 ± 0.56** |
| | 100 | - | - | - | 49.7 ± 0.3 | **50.16 ± 0.21** |

**Results on Tiny-ImageNet.** We also evaluate ours on the large-scale datasets, where we consider Tiny-ImageNet following SRe$^2$L by Yin et al. (2023) and adopt the same backbone. Since ours is so flexible that it can be combined with other methods, we consider combine it with SRe$^2$L, denoted as SRe$^2$L+OURS. As shown in Table 2, we can find that introducing ours can obtain consistent boosts on performance. That is to say, matching energy distance between real and synthetic datasets in ours is complementary to matching the statistics between real and synthetic datasets. It demonstrates that ours is still effective even in large-scale dataset with more diverse classes and larger backbone, proving its generalization and effectiveness.

**Computational Efficiency.** Considering the importance of training cost when optimizing the condensed dataset, without loss of generality, we evaluated the training speed and GPU memory required by our method and several strong baselines across different datasets under identical distillation settings. As reported in Table 3, although trajectory matching based methods usually achieve better results than distribution matching, they encounter out of memory (OOM) issues at IPC = 50. Besides, ours introduce the modest increase in computation time than NCFM, which is acceptable for maintaining competitive efficiency, especially compared with bi-level trajectory matching methods. This indicates that our method strikes a practical balance between effectiveness and computational cost.

**Learning Larger Synthetic Sets.** Prior works mainly evaluate the condensed dataset of sizes up to 50 IPCs, such as 1% compression ratio for CIFAR10, which is a rather extreme case. To get a more informative subset in most practical applications, we further explore the performance of condensation algorithms under larger compression ratios on CIFAR10, where we set IPC up to 1,000. As shown in Figure 1, the performance gain of ours over baselines is obvious for IPCs larger than 100 and less than 500. When IPC > 500, the performance gap between data condensation methods

Table 3: Comparison of training speed (s/iter) and peak GPU memory (GB) on CIFAR-100 with a single NVIDIA A100 80G. OOM indicates out-of-memory cases.

| Resource | Speed (s/iter) | | GPU Memory (GB) | |
|---|---|---|---|---|
| | IPC 10 | IPC 50 | IPC 10 | IPC 50 |
| MTT | 1.92 | OOM | 61.6 | OOM |
| FTD | 1.68 | OOM | 61.4 | OOM |
| TESLA | 5.71 | 28.24 | 10.3 | 44.2 |
| NCFM | 1.33 | 1.36 | 1.6 | 2.0 |
| NCFM+OURS | 2.15 | 2.68 | 2.1 | 3.3 |

and the random baseline narrows, where ours still outperforms other competitors. Especially, our method needs only about 500 images for each class to reach the performance of baselines trained for IPC = 1000. All methods achieve similar performance to the random baseline when IPC=1000. The possible reason is that randomly selecting more samples will approach the whole dataset, which we consider as the upper-bound. In summary, our method can be successfully applied into realistic settings, where the synthetic data of different classes can be learned independently and in parallel for avoiding bi-level optimization and second-order derivative.

Table 4: Ablation study on CIFAR10, where we only perform IPC=10,50,100 for diversity-based loss due to its degraded performance.

| Method | 10 | 50 | 100 | 200 | 300 |
|---|---|---|---|---|---|
| DM | 47.64±0.55 | 61.99±0.33 | 65.12±0.40 | 69.15±0.17 | 69.36±0.35 |
| DM + KL | 48.43±0.28 | 62.39±0.28 | 66.37±0.18 | 70.29±0.31 | 70.56±0.36 |
| DM + CLIP | 50.25±0.23 | 63.07±0.05 | 66.30±0.15 | 71.26±0.26 | 71.83±0.42 |
| DM + Classifier | 50.34±0.01 | 64.79±0.18 | 69.44±0.23 | 72.45±0.43 | 73.54±0.32 |
| Diversity | 31.02 ± 0.74 | 41.06± 0.29 | 55.61 ± 0.37 | - | - |
| Diversity + Classifier | 39.91 ± 0.51 | 48.58± 0.48 | 62.98 ± 0.33 | - | - |
| ED | 50.71±0.30 | 63.30±0.33 | 67.88±0.25 | 71.92±0.31 | 73.85±0.34 |
| ED + KL | 51.12±0.25 | 65.19±0.12 | 68.43±0.02 | 72.88±0.25 | 74.01±0.37 |
| ED + CLIP | 50.34±0.22 | 64.02±0.24 | 69.01±0.29 | 72.27±0.63 | 75.11±0.33 |
| ED + Classifier | **51.34±0.24** | **65.92±0.30** | **69.47±0.30** | **73.73±0.21** | **76.02±0.26** |

## 6.3 Ablation Studies

**Energy distance.** To illustrate the effectiveness of our introduced energy distance, we compare it with another distribution matching method, *i.e.* DM (Zhao & Bilen, 2023). Here we discard our proposed constraint for a fair comparison and denote ED as our decayed variant. As listed in Table 4, ED produces superior performance than that of DM. Especially, ED outperforms DM by large margins with larger IPCs. It is reasonable since ED not only minimizes the distance between synthetic data and whole-dataset, but also maximizes the diversity of synthetic samples. Besides, we also discard the first term (i.e., matching term between real dataset and synthetic dataset) in energy distance and we report their performance for IPC=10, 50,100. We can find that discarding the matching part can reduce the performance a lot. This is because that the matching term guides the synthetic samples toward the real data distribution.

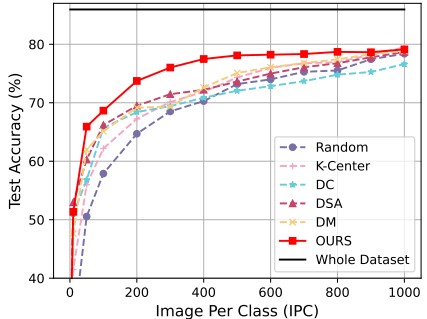

Figure 1: Performance comparison on CIFAR-10 under different IPCs using Random, K-Center, DC, DSA, DM and Ours.

When discarding the matching term, the resultant loss only pushes the synthetic samples to move away from each other in each class (via the repulsion term), without providing direction toward the target distribution. These observations suggest that using the energy distance to match feature distributions of the synthetic and original training images is effective in dataset condensation.

**Classifier-critic constraint.** To evaluate the effectiveness of our proposed classifier-critic constraint for training the synthetic images, we consider different ways to build the pre-trained classifier, including traditional classifier trained on original training set and CLIP. Besides, we also implement the constraint following knowledge distillation (Hinton et al.) by minimizing the KL divergence $\mathrm{KL}\left(p_\phi(\boldsymbol{y}|B_c^{\mathcal{T}})\|p_\phi(\boldsymbol{y}|B_c^{\mathcal{S}})\right)$, where we keep the mini-batch size for real data in accord with the size of synthetic set. As listed in Table 4, adopting the CLIP, which is trained on a public dataset instead of the target large-scale training set, is inferior to using the traditional classifier. The reason behind this might be that the traditional classifier is optimized by the original training set. However, replacing the traditional classifier with CLIP only reduce the performance a little. Compared with ignoring the classification constraint, introducing CLIP to implement the classifier-critic can result in a remarkable performance improvement when IPC is equal or greater than 50, where DM+CLIP is better than DM, ED+CLIP is better than ED, and ED+CLIP still outperforms DM+CLIP. Thus, even using existing available classifiers trained on different dataset, such as CLIP model, ours can still achieve a desired performance. Besides, KL is inferior to our method. The possible reason is that KL provides the soft label predicated by classifier $g_\phi$ on the real dataset as the target of the synthetic set but we use the ground-truth label. To sum up, introducing KL, CLIP or traditional classifier as the constraint can usually result in a remarkable performance improvement. Thus, we claim that the classifier-critic con-

straint on synthetic images is a key-point in data condensation settings; this highlights the contribution and originality of our method.

Table 5: Ablation study of hyperparameter $\lambda$ on CIFAR10 with IPC=10 and IPC=50.

| $\lambda$ | 10 | 1 | 0.1 | 0.01 | $\frac{\text{maxiter}-t}{\text{maxiter}}$ | 0 |
|---|---|---|---|---|---|---|
| IPC=10 | 51.26±0.38 | 50.30±0.54 | 50.80±0.48 | 50.17±0.26 | **53.42±0.17** | 50.71±0.30 |
| IPC=50 | **66.72±0.24** | 65.48±0.49 | 63.90±0.17 | 63.59±0.22 | 65.92±0.30 | 63.30±0.33 |

**Hyperparameter $\lambda$.** The hyperparameter $\lambda$ in Eq. (7) indicates the importance of the log-likelihood of the task-related information in synthetic set, *i.e.* the weight of task-related loss. To explore the effect of $\lambda$ in our method, we design ablation study on CIFAR10 with IPC=10/50. As listed in Table 5, a small $\lambda$, such as 0.01, achieves slightly better or even worse performance than abandoning the task-related loss *i.e.* $\lambda = 0$. Both settings of $\lambda = 10$ and $\lambda = \frac{\text{maxiter}-t}{\text{maxiter}}$ can produce a better performance than others. Since we aim to use a much more general hyperparameter rather than searching a perfect $\lambda$ for each dataset with each IPC jadedly, we set $\lambda = \frac{\text{maxiter}-t}{\text{maxiter}}$. That is to say, we wish to learn synthetic set with more emphasis on $p(\mathcal{C}|\mathcal{S})$ during the early training stage and pay more attention on $p(\mathcal{S})$ during the late training stage. Our proposed method achieves desired performance in various tasks using this hyperparameter varying with the training iterations.

**The efficiency of the CE loss during the training.** Recall that the coefficient of the additional CE loss is set to be gradually reduced during the training. To further explore the efficiency of the CE loss during the whole training process, we compare the test performance learned by ours and DM with varying number of training iterations, where we consider ours with CE loss and ours without CE loss. As shown in Figure 2, we evaluate the performance of the condensed samples at each 2000 training iterations. We can find that ours with CE loss performs better than DM in the whole training stage. Besides, ours with CE loss outperforms ours without CE loss all the time. Therefore, CE loss is not only beneficial in the later phase of the training but also useful in the beginning phase.

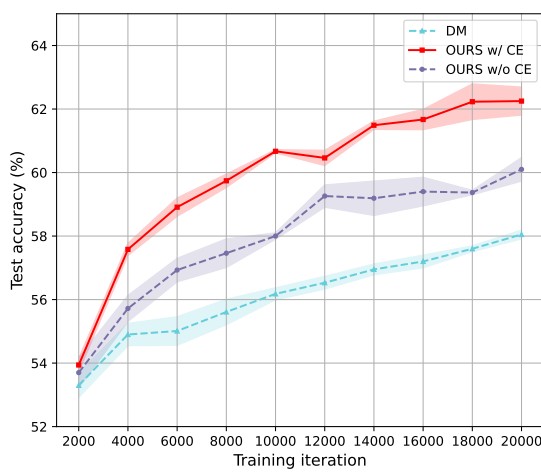

Figure 2: The test performance of ours (w/ CE loss or w/o CE loss) and DM on CIFAR10 (IPC=50) with the varying number of training iterations.

### 6.4 Transferability

Table 6: Testing accuracy of different methods on ConvNet versus transferred to other architectures, where IPC=10/50. The "Transfer" column records the average results on MLP, ResNet18, ResNet152 and ViT.

| Dataset | CIFAR-10 | | | | CIFAR-100 | | | | TinyImageNet | | | |
|---|---|---|---|---|---|---|---|---|---|---|---|---|
| Method | IPC=50 | | IPC=10 | | IPC=50 | | IPC=10 | | IPC=50 | | IPC=10 | |
| | ConvNet | Transfer | ConvNet | Transfer | ConvNet | Transfer | ConvNet | Transfer | ConvNet | Transfer | ConvNet | Transfer |
| Random | 50.55 | 36.39 | 31.00 | 24.16 | 34.66 | 23.12 | 18.64 | 11.31 | 18.62 | 8.85 | 6.88 | 3.53 |
| K-Center | 56.00 | 41.45 | 41.19 | 31.01 | 38.64 | 26.17 | 25.04 | 15.31 | 22.02 | 10.77 | 11.38 | 5.42 |
| DC | 56.81 | 31.42 | 50.99 | **32.22** | 30.56 | 14.95 | 28.42 | 11.95 | 12.66 | 4.62 | 12.83 | 3.74 |
| DSA | 60.28 | 38.37 | 52.96 | 31.15 | 43.13 | **27.11** | 32.23 | 15.77 | 25.31 | 10.93 | 16.34 | 6.75 |
| DM | 61.99 | 40.04 | 47.64 | 30.66 | 42.32 | 25.22 | 29.23 | 13.59 | 22.76 | 9.75 | 13.51 | 4.08 |
| OURS | **65.92** | **41.55** | **53.42** | 31.74 | **44.90** | 26.83 | **32.80** | **16.22** | **28.10** | **23.24** | **16.41** | **7.13** |

Following (Cui et al., 2022), we explore the transfer-
ability of different condensation methods on the 3 datasets using 5 architectures, including ConvNet, MLP, ResNet18 (He et al., 2016), ResNet152 (He et al., 2016) and ViT (Dosovitskiy et al., 2020). More details about architectures and transfer results on each architecture can be found in the Appendix B. As shown in Table 6, although the performance of all methods drops when transferring to other architectures, ours still produces a desired balance between the seen ConvNet and unseen architectures. When IPC=10 on CIFAR10

and IPC=50 on CIFAR100, ours is slightly worse than DC or DSA. In addition to above-mentioned setings, ours achieves the better performance over the seen ConvNet and better-averaged generalization performance over the unseen architectures. Remarkably, when IPC=50 on TinyImageNet, the performance gain of ours over DSA is about 14%. The possible reason is that learning the synthetic data from TinyImageNet is very challenging and ours extract more class-relevant information with less architectural inductive bias than other baselines; see the visualization below. Thus, our proposed method is reliable and generalizable whose condensed images using one architecture can be effectively used to train other unseen ones.

## 6.5 Neural Architecture Search (NAS)

NAS aims to automatically search for a top archi-tecture from a vast search space, which typically re-quires expensive training of numerous architectures multiple times on the whole training set and pick-ing the best performing ones on a validation set. It is beneficial that applying the smaller condensed dataset to NAS (Zhao et al., 2021). Following the implementations of (Cui et al., 2022), we randomly

Table 7: We implement NAS on CIFAR10 using NAS-Bench-201. The correlation of the original dataset is lower than 1.0 as we use a small architecture and per-form ranking based on the validation set.

|  | Random | K-Center | DC | DSA | DM | OURS | Whole |
|---|---|---|---|---|---|---|---|
| Correlation ↑ | -0.06 | **0.11** | -0.19 | -0.37 | -0.37 | -0.17 | 0.7487 |
| Top 1 (%) ↑ | 91.9 | 91.78 | 86.44 | 73.54 | 92.16 | **92.47** | 93.5 |

sample 100 networks from NAS-Bench-201 (Dong & Yang, 2020), which contains the ground-truth perfor-mance of 15,625 networks. We reduce the number of repeated blocks from 15 to 3 during the search phase, making the size of condensed dataset adapt the networks. All models are trained on CIFAR10 and IPC=50 for 50 epochs under 5 random seeds and ranked according to their average accuracy on a held-out validation set of 10k images. We consider two metrics 1) Spearman's rank correlation coefficient between the ranking of models trained on condensed dataset and the original dataset; 2) The ground-truth performance of the best architecture trained on the condensed dataset (Top 1). As we can see in Table 7, all methods produce negative correlation between the performance on condensed and full dataset except for K-Center, indicating the difficulty of utilizing the condensed dataset to guide model designs. Besides, ours still achieves a rel-atively better trade-off between correlation and ground-truth performance than other methods. It verifies that our condensed images can be used to efficiently train multiple networks to identify the best network for saving computation resources.

## 6.6 Continual Learning (CL)

CL (Kirkpatrick et al., 2017) aims to address the catastrophic forgetting problem when the model learns sequentially from a stream of tasks. Due to its highly condensed nature, distilled samples have been success-fully applied to the CL. We evaluate the effectiveness of the condensed dataset in CL. Following (Zhao & Bilen, 2023), we set up the baseline based on GDumb (Prabhu et al., 2020) which stores class-balanced train-ing samples in memory greedily and trains a model from scratch on the latest memory only. We randomly and evenly split the 100 classes from CIFAR100 dataset into 5 steps, *i.e.* 20 classes per step respectively. The memory budget is 20 images/class for all seen classes. We compare our method with other baselines. Figure 3 shows that the proposed method is superior to other baselines. It indicates that our method can pro-duce a better and more informative condensed set for training models than those produced by competitors. Similarly, the results of 10 step class-incremental learning are shown in Figure 4.

## 6.7 Visualization of Data Distribution

Considering data condensation methods synthesize a small dataset, one natural question is whether it intro-duces any bias into the data distribution. Therefore, we visualize the data distribution of the real dataset and the synthetic datasets by different methods in Figure 5 for CIFAR10 with IPC=50. Specifically, we use the ResNet18 pretrained on the real dataset to extract features and visualize the features with T-SNE (Van der Maaten & Hinton, 2008), where we plot the samples from the first class, second class and fourth class. We find that the synthetic images learned by baselines are usually biased to real image distribution, especially for DC and CAFE. However, ours can obtain more evenly distributed samples and cover the real

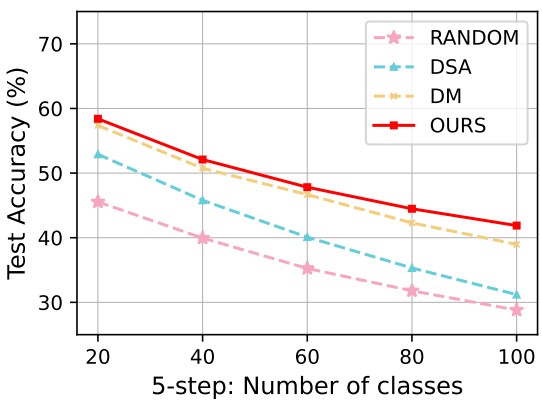

Figure 3: Test accuracy of 5 steps continual learning with condensed samples on CIFAR100 with IPC=20 for Random, DM, DSA and ours, respectively.

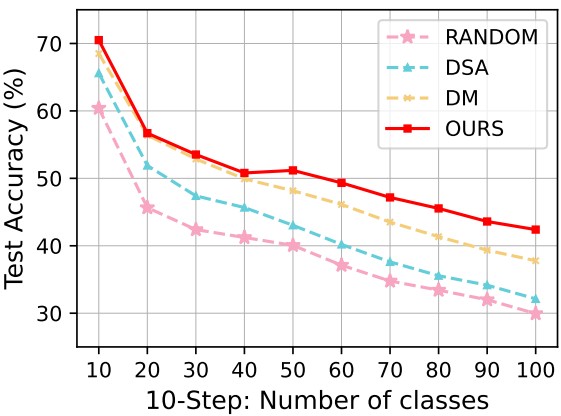

Figure 4: Test accuracy of 10 steps continual learning with condensed samples on CIFAR100 with IPC=20 for Random, DM, DSA and ours, respectively.

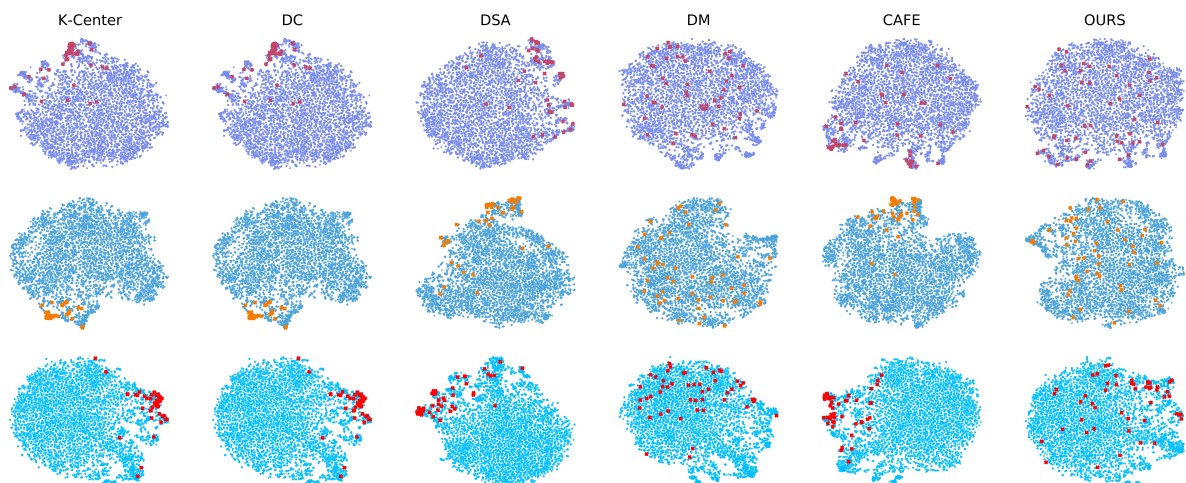

Figure 5: Visualization of synthetic dataset distribution (red points) and original dataset (blue points) on CIFAR10 with IPC=50, where the first row is the first class, and the second row is the second class and the third row is the fourth class.

image distribution in a better way. It thus proves the effectiveness of ours in above-mentioned downstream tasks.

## 6.8 Debiased Dataset Condensation

Table 8: Performance on CelebA with different IPCs. The average and worst performance on the whole dataset are $94.53 \pm 0.19$ and $93.85 \pm 0.27$, respectively.

| Method | 10 | | 50 | | 100 | | 200 | | 300 | |
|---|---|---|---|---|---|---|---|---|---|---|
| | Avg. | Worst | Avg. | Worst | Avg. | Worst | Avg. | Worst | Avg. | Worst |
| RANDOM | 80.30±0.29 | 76.18±0.29 | 86.68±0.13 | 77.44±0.41 | 83.45±0.23 | 81.69±0.57 | 88.05±0.09 | 84.66±0.66 | 88.29±0.26 | 86.37±0.12 |
| DM | 86.68±5.31 | 66.71±2.64 | 87.56±1.48 | 84.44±1.44 | 88.51±0.56 | 87.89±0.04 | 89.74±0.40 | 88.64±0.28 | 89.40±0.36 | 88.36±0.38 |
| ED | 86.48±1.17 | 69.42±1.48 | 86.86±0.75 | 84.63±0.36 | 88.86±1.07 | 88.38±0.88 | 90.61±0.56 | 89.60±0.30 | 90.72±0.05 | 89.56±0.90 |
| ED+label | **86.76±1.49** | 75.64±0.95 | 87.85±1.07 | 86.48±1.68 | 89.31±0.25 | 87.04±0.80 | 90.33±0.31 | 89.66±0.25 | 91.09±0.32 | 89.84±0.13 |
| OURS | 86.66±1.38 | **76.78±1.68** | **87.85±0.06** | **86.48±0.48** | **89.54±0.56** | **88.38±0.26** | **90.71±0.63** | **89.71±0.66** | **91.48±0.38** | **90.74±0.09** |

We examine whether ours can be used to compress a biased training set into a small de-biased synthetic dataset. Here we adopt the **CelebA** face dataset (Liu et al., 2015) that includes 40 attributes. We use the hair color ($blond, not-blond$) as the target and gender ($male, female$) as the spurious attribute, where the label is spuriously correlated with gender. We now have 4 groups: (blond, male), (not blond, female), (blond, female), (not blond, male). To reduce the computational resource, we resize the original image into $64 \times 64$ and keep the proportions of 4 groups in original dataset to randomly drop some samples, resulting in 10000 training examples with 85 in the smallest group (blond-haired males). See Appendix C for more details. We adopt the official test split to evaluate the methods and report the average and worst-group performance. We follow (Graikos et al., 2022) that uses the pretrained ResNet18 face attribute classifier on CelebA to realize both $\phi_1$ and $\phi_2$. We consider random selection, DM, ED, and ED+label as our baselines, where ED+label adds the minimization about the CE loss of label in the distilled image to the ED. Ours further adds the maximization about the CE loss of spurious attribute to the ED+label. As listed in Table 8, ours can generally achieve the better worst-group and average performance than its baselines, indicating the effectiveness of introduced constraints in de-biased dataset condensation. Besides, adding the constraint about the label usually leads to improved average performance. It is reasonable since ED+label pays more attention to all groups. These observations claim that introducing classifier-critic constraint about the specific task is effective in dataset condensation and can provide more applications. We also perform a data condensation on point cloud classification task using our proposed method and report the results in Appendix 7, to validate the effectiveness of ours on different modalities.

## 6.9 Visualization of Condensed Dataset

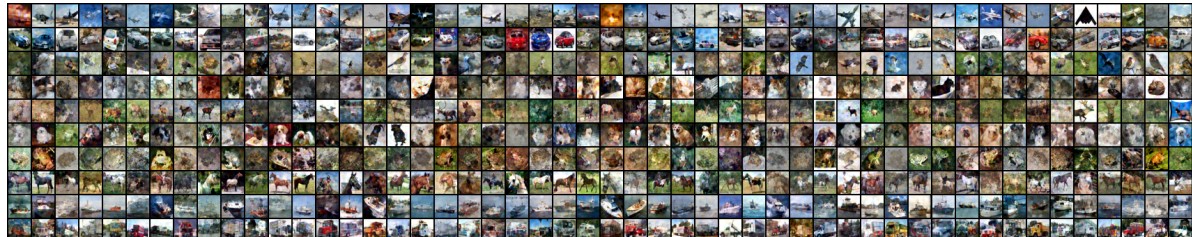

Figure 6: Visualization of synthetic images on CIFAR10 with IPC=50. Please zoom in for a better display.

We visualize the synthetic images on CIFAR10 with IPC=50 in Figure 6. Due to the limited space, we defer the generated synthetic sets of CIFAR10 with IPC=10 in Figure 7 of Appendix and the generated synthetic sets of CelebA with IPC=10 in Figure 8 of Appendix, respectively. With the development of IPC, it is easy to find that the synthetic images generated by our method is visually similar to original real CIFAR10 images but looks more class-representative. Taking the Figure 6 (the visualization of synthetic images CIFAR10 with IPC=50) as the example, the 50 images per class results are diverse which cover the main variations, such as the various airplanes in the first row ("airplane" class). For CelebA, we consider the debiased dataset condensation, where the hair color (blond or not-blond) is the label and the gender (male or female) is the spurious attributes. That is to say, we aim to obtain a condensed face dataset with two classes, where one class is composed of blond hair face images and another is composed of non-blond hair face images and both of them can not keep the gender information in the images as soon as possible. As shown in Figure 8 of Appendix, we can see that the condensed can meet the expected constraints to a great extent.

## 7 Results on Other Modalities

To validate the effectiveness of ours on different modalities, we perform a data condensation on point cloud classification task using our proposed method. Here we use preprocessed modelnet10 dataset (Wu et al., 2015), which contains 3981 training and 400 testing samples from 10 categories and each sample is composed of 10K points (x;y; z-coordinates;R;G;B). Researchers usually utilize random subsampling to achieve fewer points (e.g., 2048) for each object. We compare random sampling and DM, our proposed ED, and ED+classifier to achieve fewer points. We randomly sample 100 training samples and condense each sample with 10K 3D points (x;y; z-coordinates) into a synthetic sample with 64 3D points. We then utilize condensed

datasets to train the pointnet++ for point cloud classification task, where we set epoch=50. We adopt a common evaluation method for dataset condensation methods, i.e. measuring the test accuracy of the neural networks trained on the condensed data, where we use all test samples and report instance accuracy (IA) and class accuracy (CA). The test performance is summarized in Table 9. Compared with random sampling and DM, ours can generate more data-efficient samples, indicating the effectiveness of our proposed data condensation method in 3D.

Table 9: The accuracy comparison of IA and CA for point cloud classification.

| Method | Random | ED | DM | ED+classifier |
|--------|--------|-------|-------|---------------|
| IA | 51.92 | 62.87 | 62.20 | **67.24** |
| CA | 53.43 | 65.01 | 64.28 | **68.93** |

## 8 Conclusion

This work aims to compress the original training dataset into a condensed set via class-preserving distribution matching. To this end, we design the optimization method of synthetic samples from two key points. One is responsible for capturing the original data distribution, where we introduce energy distance to increase the diversity. The other aims to introduce the task-related information, such as classification, into the learning of synthetic samples without a bi-level optimization. We introduce a classifier-critic constraint based on an off-the-shelf pre-trained classifier to implement the second goal. By minimizing the energy distance between real and synthetic samples and the classifier-critic constraint, we learn a diverse and class-preserving synthetic set with less training cost, avoiding an expensive bi-level optimization. Extensive experiments demonstrate the effectiveness of our proposed method in commonly used downstream tasks and the de-biased dataset condensation task.

## 9 Acknowledgment

This work was primarily conducted while Dandan Guo was a postdoctoral researcher at The Chinese University of Hong Kong, Shenzhen. The work of Dandan Guo was supported by the National Natural Science Foundation of China (NSFC) under Grant 62306125. This work of Hongyuan Zha was supported by the National Natural Science Foundation of China (NSFC) under Grant 72495131.

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

## A   Detailed Implementation Settings

We list our used learning rate for each dataset with varying IPCs in Table 10 and the mini-batch size of the real dataset in Table. 11. We also report the pretraining details for the model parameters $\phi$ on different datasets in Table 12. For a standard dataset condensation task, $\phi$ indicates a classifier and its training details can be found in 12, where we adopt the same architecture and settings with the downstream classification task. For the di-biased dataset condenstaion task, $\phi$ represents a face attribute classifier, we use a well pretrained model following (Graikos et al., 2022).

Table 10: Learning rate on CIFAR-10, CIFAR-100, TinyImageNet and CelebA with different IPCs. To obtain the results in Figure 1, we set the learning rate as 6 with IPC $\geq$ 300 on CIFAR-10.

| IPC | CIFAR-10 | CIFAR-100 | TinyImageNet | CelebA |
|---|---|---|---|---|
| 1 | 0.1 | 0.1 | 0.2 | - |
| 10 | 0.1 | 0.1 | 0.2 | 10 |
| 50 | 0.1 | 0.1 | 0.2 | 10 |
| 100 | 1 | - | - | 10 |
| 200 | 1 | - | - | 10 |
| 300 | 5 | - | - | 10 |

Table 11: Mini-batch size of real dataset on CIFAR-10, CIFAR-100 and TinyImageNet with different IPCs.

| Dataset | IPC | | | | | | | | | | | | |
|---|---|---|---|---|---|---|---|---|---|---|---|---|---|
| | 1 | 10 | 50 | 100 | 200 | 300 | 400 | 500 | 600 | 700 | 800 | 900 | 1000 |
| CIFAR-10 | 512 | 512 | 512 | 512 | 512 | 512 | 512 | 512 | 256 | 256 | 64 | 64 | 64 |
| CIFAR-100 | 512 | 512 | 512 | - | - | - | - | - | - | - | - | - | - |
| TinyImageNet | 512 | 512 | 256 | - | - | - | - | - | - | - | - | - | - |
| CelebA | - | 512 | 512 | 512 | 512 | 512 | - | - | - | - | - | - | - |

Table 12: Training details when $\phi$ indicates a classifier on CIFAR-10, CIFAR-100 and TinyImageNet, where BS denotes Batch Size, E denotes Epoch, lR denotes Learning rate, Optim denotes Optimizer, Mome denotes Momentum and WD denotes Weight Decay.

| Data | Model | BS | E | LR | Scheduler | Optim | Mome | WD |
|---|---|---|---|---|---|---|---|---|
| CIFAR10 | ConvNet | 256 | 1000 | 0.01 | STEPLR | SGD | 0.9 | 5e-4 |
| CIFAR100 | ConvNet | 256 | 1000 | 0.01 | STEPLR | SGD | 0.9 | 5e-4 |
| TinyImageNet | ConvNetD4 | 32 | 1000 | 0.01 | STEPLR | SGD | 0.9 | 5e-4 |
| CelebA | ResNet18 | 256 | 1000 | 0.01 | STEPLR | SGD | 0.9 | 5e-4 |

## B   More Details about Transfer Experiments

Following (Cui et al., 2022), the evaluated network architectures are as follows:

**ConvNet:** A standard convolutional network which is classical in vision tasks, is used to extract features from the synthetic and original datasets in previous works (Zhao & Bilen, 2023). The structure includes three 3×3 convolutional layers followed by a 2×2 avgpooling and instance normalization with hidden widths of 128 on CIFAR-10 and CIFAR-100. For TinyImagenet, a four layers network called ConvNetD4 is used for better performance and ability in extracting features following (Cui et al., 2022). There are 0.32M trainable parameters in ConvNet and 0.45M in ConvNetD4, correspondingly.

**MLP:** A simple fully connected network with 3 layers and 128 neurons in each layer is considered to evaluate methods. There are 0.41M trainable parameters in MLP.

**ResNet18/ResNet152:** Standrad architecture includes 4/50 residual blocks, resepectively. A ReLU activation layer and batch normalization layer in each block follow two convolution layers. There are 11.17M trainable parameters in ResNet18 and 58.16M in ResNet152.

**ViT:** Vision Transformer is a new and competitive model architecture used in image processing, which is based on transformer structure. There are 10M trainable parameters in our ViT.

Previously, we show the transfer results in Table. 6 with IPC=50. We further report the detailed transfer results on each architecture in Table. 14 for easier references.

## C  Details about the De-biased Dataset Condensation

We extract features and evaluate the synthetic images in ResNet18, where we use the same architecture in transfer experiments. Refer to Table. 13 for details about dataset.

Table 13: Number of samples in different groups in training dataset and test dataset on CelebA.

| Group | Blond Male | Not blond Male | Blond Famale | Not blond Famale | All |
|---|---|---|---|---|---|
| Original training dataset | 1387 | 66874 | 22880 | 71629 | 162770 |
| Our training dataset | 85 | 4101 | 1405 | 4401 | 10000 |
| Test dataset | 180 | 2480 | 7536 | 9767 | 19963 |

## D  Visualization of the Learned Synthetic Samples

- We visualize the generated synthetic sets of CIFAR10 with IPC=10 in Figure7, respectively.

- We visualize the generated synthetic sets of CelebA with IPC=10 in Figure8 , respectively.

We can observe that the synthetic images can capture diverse appearances in the categories. Besides, the synthetic images in each class are less redundant.

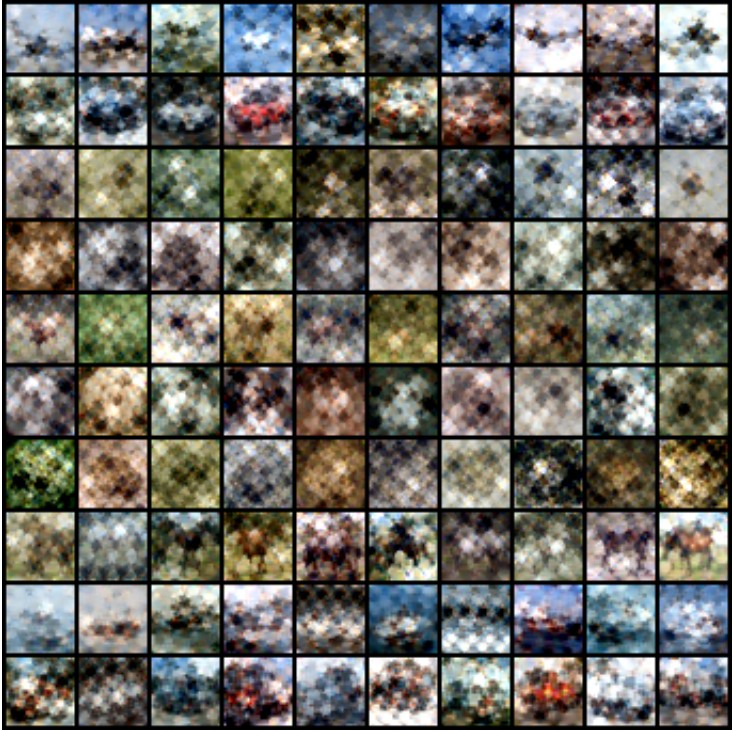

Figure 7: Visualization of synthetic images on CIFAR10 with IPC=10.

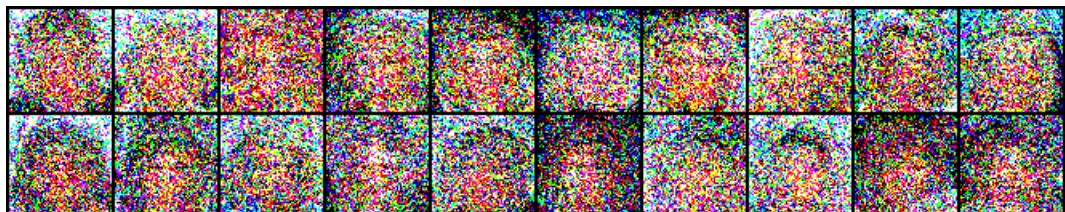

Figure 8: Visualization of synthetic images on CelebA with IPC=10. Please zoom in for a better display.

Table 14: Comprehensive transferability results of different methods tested on ConvNet, MLP, ResNet18, ResNet152 and ViT with IPC 1, 10 and 50. We report testing accuracy.

| Dataset | Method | IPC | ConvNet | MLP | Network ResNet18 | ResNet152 | ViT |
|---|---|---|---|---|---|---|---|
| CIFAR-10 | Random | 1 | 15.40±0.28 | 14.37±0.38 | 16.56±0.46 | 12.15±1.80 | 14.19±0.99 |
| | | 10 | 31.00±0.48 | 25.08±0.27 | 29.52±0.87 | 15.84±0.91 | 26.21±0.49 |
| | | 50 | 50.55±0.32 | 25.21±0.44 | 47.26±0.27 | 23 36±2.31 | 39.73±0.52 |
| | K-Center | 1 | 25.16±0.45 | 24.01±0.32 | 25.99±0.57 | 14.64±1.30 | 21.54±0.55 |
| | | 10 | 41.49±0.73 | 32.92±0.38 | 40.08±0.88 | 19.35±0.71 | 31.95±0.57 |
| | | 50 | 56.00±0.29 | 40.61±0.34 | 52.69±0.70 | 27.84±1.07 | 44.65±0.39 |
| | DC | 1 | 29.34±0.37 | 29.02±0.52 | 27.43±0.71 | 15.31±0.36 | 28.14±1.11 |
| | | 10 | 50.99±0.62 | 34.06±0.40 | 43.96±1.37 | 16.51±0.89 | 34.36±0.35 |
| | | 50 | 56.81±0.44 | 31.63±0.55 | 45.94±1.41 | 17.98±1.06 | 30.14±0.51 |
| | DSA | 1 | 27.76±0.47 | 25.04±0.77 | 25.59±0.56 | 15.12±0.65 | 23.70±0.20 |
| | | 10 | 52.96±0.41 | 34.49±0.47 | 42.11±0.56 | 16.10±1.03 | 31.88±0.35 |
| | | 50 | 60.28±0.37 | 41.01±0.36 | 49.52±0.72 | 19.65±1.16 | 43.30±0.43 |
| | DM | 1 | 26.45±0.39 | 10.02±0.55 | 20.64±0.47 | 14.09±0.58 | 20.47±0.46 |
| | | 10 | 47.64±0.55 | 34.44±0.30 | 38.21±1.05 | 15.60±1.51 | 34.37±0.49 |
| | | 50 | 61.99±0.33 | 40.49±0.38 | 52.76±0.44 | 21.67±1.34 | 45.22±0.37 |
| | CAFE | 1 | - | - | - | - | - |
| | | 10 | 50.68±0.17 | 35.94±0.65 | 42.40±0.88 | 25.51±0.41 | 35.22±0.16 |
| | | 50 | 61.97±0.22 | 24.41±0.17 | 26.12±0.77 | 14.78±0.55 | 25.10±0.37 |
| | OURS | 1 | 29.30±0.33 | 24.58±0.10 | 24.56±0.34 | 13.60±0.21 | 20.74±0.30 |
| | | 10 | 53.42±0.14 | 34.94±0.36 | 40.65±0.14 | 17.20±0.33 | 34.17±0.16 |
| | | 50 | 65.92±0.24 | 41.40±0.67 | 55.49±0.21 | 23.12±0.42 | 46.20±0.29 |
| CIFAR-100 | Random | 1 | 5.30±0.23 | 4.27±0.09 | 4.36±0.15 | 1.73±0.12 | 4.45±0.15 |
| | | 10 | 18.64±0.25 | 10.20±0.18 | 15.77±0.24 | 5.19±0.46 | 14.07±0.21 |
| | | 50 | 34.66±0.41 | 16.80±0.31 | 30.23±0.61 | 18.55±1.29 | 26.90±0.33 |
| | K-Center | 1 | 10.89±0.17 | 7.96±0.17 | 8.75±0.43 | 2.22±0.19 | 7.81±0.13 |
| | | 10 | 25.04±0.30 | 13.92±0.20 | 22.18±0.59 | 7.14±0.79 | 17.98±0.44 |
| | | 50 | 38.64±0.43 | 19.32±0.36 | 34.00±0.51 | 21.25±1.46 | 30.12±0.65 |
| | DC | 1 | 13.66±0.29 | 9.78±0.27 | 9.71±0.46 | 2.67±0.16 | 9.27±0.14 |
| | | 10 | 28.42±0.29 | 12.36±0.20 | 17.94±0.59 | 5.28±1.05 | 12.22±0.17 |
| | | 50 | 30.56±0.56 | 13.29±0.30 | 17.64±0.31 | 11.36±0.95 | 17.51±0.15 |
| | DSA | 1 | 13.73±0.45 | 10.56±0.22 | 9.95±0.55 | 2.95±0.44 | 9.48±0.27 |
| | | 10 | 32.23±0.35 | 16.17±0.26 | 21.86±0.43 | 5.45±1.04 | 19.61±0.15 |
| | | 50 | 43.13±0.33 | 21.42±0.31 | 34.34±0.44 | 20.79±1.76 | 31.89±0.49 |
| | DM | 1 | 11.20±0.27 | 8.17±0.21 | 5.36±0.31 | 2.11±0.13 | 4.59±0.26 |
| | | 10 | 29.23±0.26 | 14.68±0.18 | 18.72±0.49 | 3.91±0.73 | 17.06±0.25 |
| | | 50 | 42.32±0.37 | 20.14±0.24 | 33.34±0.40 | 17.29±2.41 | 30.11±0.25 |
| | OURS | 1 | 12.80±0.20 | 6.71±0.16 | 5.76±0.11 | 2.26±0.32 | 7.28±0.19 |
| | | 10 | 33.39±0.22 | 16.06±0.08 | 22.58±0.33 | 6.73±0.17 | 19.51±0.23 |
| | | 50 | 45.19±0.18 | 19.63±0.17 | 38.54±0.66 | 18.44±0.36 | 30.70±0.21 |
| TinyImageNet | Random | 1 | 1.65±0.11 | 1.37±0.08 | 1.27±0.08 | 0.63±0.08 | 1.71±0.03 |
| | | 10 | 6.88±0.25 | 3.12±0.13 | 3.34±0.16 | 1.01±0.15 | 6.63±0.21 |
| | | 50 | 18.62±0.22 | 5.28±0.2 | 10.35±0.33 | 2.90±0.40 | 16.87±0.20 |
| | K-Center | 1 | 3.03±0.12 | 2.53±0.13 | 2.29±0.10 | 0.82±0.10 | 2.27±0.02 |
| | | 10 | 11.38±0.26 | 4.73±0.08 | 5.46±0.24 | 1.55±0.21 | 9.92±0.34 |
| | | 50 | 22.02±0.40 | 5.99±0.17 | 13.51±0.34 | 3.91±0.49 | 19.70±04 |
| | DC | 1 | 5.27±0.10 | 2.67±0.17 | 3.17±0.21 | 0.90±0.14 | 2.00±0.12 |
| | | 10 | 12.83±0.14 | 4.12±0.11 | 5.44±0.21 | 1.24±0.18 | 4.17±0.10 |
| | | 50 | 12 66±0.36 | 3.81±0.17 | 7.05±0.21 | 2.39±0.21 | 5.22±0.23 |
| | DSA | 1 | 5.67±0.14 | 3.90±0.16 | 3.20±0.13 | 0.84±0.12 | 3.17±0.03 |
| | | 10 | 16.34±0.21 | 6.31±0.21 | 7.60±0.36 | 1.90±0.21 | 11.17±0.15 |
| | | 50 | 25.31±0.22 | 6.72±0.20 | 13.36±0.40 | 3.78±0.56 | 19.87±0.44 |
| | DM | 1 | 3.82±0.21 | 3.11±0.10 | 1.79±0.17 | 0.89±0.07 | 3.21±0.07 |
| | | 10 | 13.51±0.31 | 4.24±0.13 | 3.57±0.20 | 1.06±0.19 | 7.46±0.20 |
| | | 50 | 22.76±0.28 | 5.74±0.27 | 11.07±0.39 | 3.33±0.46 | 18.88±0.36 |
| | OURS | 1 | 4.41±0.37 | 2.20±0.17 | 1.64±0.07 | 0.81±0.16 | 3.74±0.11 |
| | | 10 | 16.94±0.15 | 3.57±0.16 | 8.98±0.28 | 6.28±0.04 | 9.69±0.21 |
| | | 50 | 28.10±0.21 | 25.04±0.12 | 22.44±0.31 | 23.74±0.42 | 21.72±0.37 |

