# OpenReview forum: "Diverse Condensed Data Generation via Class Preserving Distribution Matching"
_TMLR — Accepted by TMLR_

### Review · Reviewer_om5U · 2025-04-29

**Summary Of Contributions:**

The paper studies the data condensation problem and proposes a method to provide a condensed dataset with both improved diversity and good distribution matching to the original large-scale dataset. Existing methods either lack good diversity or require bi-level optimization. The paper uses a class-preserving distribution matching approach and energy-distance-based distribution matching to solve the limitations. Meanwhile, other data quality metrics, such as debiased data, can also be enhanced by the proposed method. Experiments are provided to demonstrate its effectiveness.

**Audience:**

Yes

**Broader Impact Concerns:**

No.

**Claims And Evidence:**

Yes

**Requested Changes:**

* It would be good to include a more comprehensive discussion of the related work and experimental comparison with the baseline method.
* It would be good to have more discussion about the computational benefit to support the claim at the beginning.
* It would be good to add more discussion about the unique method design and the contribution over existing methods.

**Strengths And Weaknesses:**

Strengths:
* The paper is well-written and easy to follow.
* The paper has relatively solid experiments to show the benefit of the proposed method.
* The paper has good visualization.
* The proposed method is practical and easy to incorporate into the existing method.

Weaknesses:
* The related work and baseline methods in the experiments do not include all the important methods. Some more recent methods are not incorporated.
* The bi-level optimization has be dealt with by some existing works, like using multiple early-stopping models to reduce the computation. I was wondering what the advantages of the proposed method are. Also, it would be good to add some computational comparisons in the experiments.
* The related work regarding diverse data condensation is insufficient.
* Apart from the debiased quality, could the method also introduce other good data quality?
* The proposed method seems to simply combine two methods and lacks novelty.

---

> ### Author Response · Authors · 2025-05-27
> **Response to Reviewer om5U**
>
> (1) Response to W1, W3 and RC1: Thanks for your suggestions. We have included the comprehensive discussion of recent works including diverse data condensation in related work (please see Section 2.1), and we have reported experimental comparison with recent baseline methods (please see Section 6.2).
>
> (2) Response to W2 and RC2: Thanks for your suggestion. I have rewritten the related work in our revision. We have added the computational comparison and performance comparison with recent baselines in experiments.
>
> (3) Response to W4: Thanks for your comments. Apart from debiased quality, the method introduced in the paper could indeed contribute to other aspects of good data quality. (1)  The use of energy distance encourages the synthetic data to match the distribution of real data in a class-wise manner and keeps the diversity. Diverse synthetic data avoids redundancy and overfitting, improving generalization in downstream models. (2) The classifier-critic constraint ensures that the synthetic data is aligned with the knowledge embedded in real-world classifiers. The synthetic samples can also satisfy other conditions.  This ensures that the distilled data is not only realistic but also relevant for actual learning tasks. For example, we can enforce the synthetic dataset aligns with the semantic information embeded in the vision language model, such as CLIP. Our proposed approach likely improves task relevance, which is important in daatset distillation.
>
> (4) Response to W5 and RC3: Thanks for your suggestion. To implement dataset condensation, there is a popular research based on distribution matching (DM) loss. It is reasonable for us to further explore a better framework for DM. To this end, how to introduce a proper distance metric into the dataset condensation is a key point. It is worth noting that energy distance is a general tool for measuring distance, which can be used to solve various problems. In this work, we find that energy distance has an attractive advantage when being used to learn condensed samples, which can not only minimize the distance between synthetic samples and real samples but also maximize the distance between synthetic samples.  To the best of our knowledge, no one has introduced the energy distance to the dataset condensation. Moreover, compared with DM that only solves the DM problem with MMD but ignores the classification performance of the condensed samples, we propose a novel class-preserving DM framework by further introducing a classifier-critic constraint, which can enforce the learned samples to distill the discriminative information beneficial for the classification task. Serving as a fundamental DM loss, ours is very flexible and can be easily combined with other methods. For example, ours can serve as a regularization loss for other optimization-based loss methods, and can also assimilate the characteristic function to measure each point-wise distance in energy distance. We have added the discussion in our introduction, related work and also Section 5 in our revision.

---

### Review · Reviewer_AXfZ · 2025-05-03

**Summary Of Contributions:**

The authors propose a new dataset condensation (or distillation) method that falls into the category of distribution matching. This line of work is cheaper than the line of work that matches gradient to condense the dataset, as it avoids bi-level optimization and the second-order derivative during the optimization process. Compared to previous distribution matching work (DM), the paper introduces two main changes:

- Classifier-critic: They add a loss term that forces the synthetic sample to be classified to its corresponding label by the classifier while not relying on spurious attribute.
- Energy distance: Adds repulsive term between the generated data so that condensed samples preserve some diversity internally.

**Audience:**

Yes

**Broader Impact Concerns:**

I do not have concerns regarding ethical issues.

**Claims And Evidence:**

Yes

**Requested Changes:**

- In Table 1, the CIFAR-100 results incorrectly bold the OURS+M3D method, even though IDM achieves higher accuracy.
- If the authors intend to emphasize the efficiency of their method, it would be more convincing to include experiments using larger models (e.g., ResNet-50) or more challenging datasets (e.g., ImageNet instead of TinyImageNet). Such results would better support claims of scalability and practical relevance.

**Strengths And Weaknesses:**

**Strengths:**

- The proposed method is conceptually simple and demonstrates strong performance within a specific compression range, particularly between 10 to 800 images per class (IPC) on the CIFAR-10 dataset.

**Weakness:**
- The primary limitation of the method is its lack of robustness in extreme compression scenarios (i.e., IPC < 10 or IPC > 1000). Although it performs well within the mid-range and the authors argue that this range is more relevant for practical applications, this claim is debatable. A more practical question would be: What is the minimum compression ratio at which the method maintains nearly lossless accuracy (e.g., within 1 percentage point of the full dataset accuracy)? The proposed method offers no clear advantage in this regard, as its performance converges to that of the random baseline beyond 1000 IPC on CIFAR-10.

- In terms of training cost, the method does not Pareto-dominate the DM baseline. This raises additional concerns about its efficiency and practical applicability.

---

> ### Author Response · Authors · 2025-05-27
> **Response to Reviewer XfZ**
>
> (1)Response to W1: Thanks for your comments. We kindly remind you that, when IPC=1000, not only ours is similar to random selection  but also the baselines are similar to random selection. For regular data condensation methods, it is a common phenomenon, which has been noticed by recent studies [1-3]. The main reason can be explained as: ``In low-IPC settings, easy patterns prove the most beneficial since they explain a larger portion of the real data distribution than an equivalent number of hard samples. However, with a sufficiently large synthetic set, learning hard samples becomes optimal since their union covers both the easy and “long-tail” hard samples of the real data. Regular distillation methods default toward distilling easy patterns, leading to their ineffectiveness in high-IPC cases.''  To this end, some solutions have been proposed for lossless dataset distillation methods, which seems more like  orthogonal to regular dataset distillation. For example, [1] align the difficulty of the learned patterns with the size of the distilled dataset when matching the training trajectories; [2] propose to initialize the synthetic dataset with real images of suitable difficulty level optimized for each IPC; [3] design a curriculum framework that gradually expands the learned synthetic dataset by incorporating suitable real samples.
>
>
> Different from those lossless dataset distillation methods that aim to design the lossless distillation strategy and focus more on larger IPC settings, our proposed method and our baselines belong to the group of designing the loss function for optimizing the distilled dataset. Besides, ours and baselines can usually be further improved by the lossless dataset distillation methods, which will be our future work.
>
> [1] Ziyao Guo, Kai Wang, George Cazenavette, Hui Li, Kaipeng Zhang, and Yang You. Towards lossless dataset distillation via difficulty-aligned trajectory matching. In ICLR, 2024.
>
> [2] Yongmin Lee and Hye Won Chung. Selmatch: Effectively scaling up dataset distillation via selection-based initialization and partial updates by trajectory matching. In ICML, 2024.
>
> [3] Chen, Yanda and Chen, Gongwei and Zhang, Miao and Guan, Weili and Nie, Liqiang. Curriculum Coarse-to-Fine Selection for High-IPC Dataset Distillation. In CVPR, 2025.
>
> (2) Response to W2: Thanks for your comments. DM is an efficient method but its performance is inferior to ours with the development of training epochs. And ours has a better trade-off for efficiency and performance. We further provide a comparison between ours and competing baselines about the training speed and GPU memory in Table 3 in our revision.
>
> (3) Response to RC1: We have fixed the typo in our revision.
>
> (4) Response to RC2: Thanks for your suggestion. We further performed the experiments on the ResNet18 about TinyImageNet. Due to the limited time, we will report the results on more challenging datasets or challenging backbones in our camera ready.

---

### Review · Reviewer_gXRh · 2025-05-06

**Summary Of Contributions:**

This paper proposes a new dataset condensation method based on existing distribution matching methods. It combines an energy distance of the data with a pre-trained classifier constraint. It shows improved performance compared to other methods including DC, DSA, DM, and CAFE. The paper further demonstrates its effectiveness regarding training cost and its performance for debiased dataset condensation.

**Audience:**

No

**Claims And Evidence:**

No

**Requested Changes:**

Please address Weakness #1.

**Strengths And Weaknesses:**

### Strengths

1. The paper covers various experiments on the application of the condensed data.

2. The method is well-motivated.

### Weakness

1. The reviewer's primary concern is the lack of comparison to related work and, therefore, limited new insights for the dataset condensation community. The baselines compared in this work (DC, DSA, DM, and CAFE) were mostly published before 2023. After these initial works in dataset condensation, various papers published in 2023 pushed the frontiers of dataset condensation: including 1) Those on the line of bi-level optimization. Cazenavette et al., 2022 proposed a trajectory matching method and achieved performance better than all the numbers in the "OURS" column reported in the paper. Follow-up work aims to alleviate the computational cost with approximation (Loo et al. 2022, Loo et al. 2023) and simplified unrolling (Feng et al. 2023). It is not yet established that the proposed method could achieve better performance with the same or less computational cost compared to these papers. 2) Those on the line of distribution matching. Yin et al. 2023 proposed a similar pipeline of first training a classifier, then learning the condensed images by matching the mean and standard deviation of the normalization layer (thus including the diversity part of the energy function), and finally relabeling them with the classifier (thus minimizing the classification loss). This paper scaled up dataset condensation from small datasets like CIFAR10 or CIFAR100 for the first time to ImageNet-1K and ResNet 50, while this paper is still training with ConvNet and with Tiny-ImageNet being the largest dataset used. Yin et al. 2023 should be a necessary comparison. The reviewer did not follow the literature since these works and there could be various follow-ups, and none of these works are cited in this paper. With these highly-cited prior works that have scaled up performance, the current experiments seem providing limited new information to the community.

2. The ablation study: The current ablation study only experiments with changing the objective from DM to ED and including the classifier as regularization. Since the ED objective consists of two parts – a matching part between distributions X and Y, and the distances between samples in dataset Y, which helps preserve diversity – it would be helpful to understand which of them helps improve performance from DM to ED. How would the method perform with only the diversity term?

3. Table 8: Why is there no result with other dataset condensation methods?

4. It is recommended to include the objective for the "OURS+M3D" method in the main paper for completeness.


-----

### Reference

Loo, Noel, et al. "Efficient dataset distillation using random feature approximation." Advances in Neural Information Processing Systems 35 (2022): 13877-13891.

Loo, Noel, et al. "Dataset distillation with convexified implicit gradients." International Conference on Machine Learning. PMLR, 2023.

Feng, Yunzhen, Shanmukha Ramakrishna Vedantam, and Julia Kempe. "Embarrassingly Simple Dataset Distillation." The Twelfth International Conference on Learning Representations. 2023.

Yin, Zeyuan, Eric Xing, and Zhiqiang Shen. "Squeeze, recover and relabel: Dataset condensation at imagenet scale from a new perspective." Advances in Neural Information Processing Systems 36 (2023): 73582-73603.

---

> ### Author Response · Authors · 2025-05-27
> **Response to Reviewer gXRh**
>
> Response to W1: Thank you for your valuable suggestion. In our revision, we have cited these work and discussed them in the related work. Besides, we have compared ours with these baselines in the experiments. Both of SRe$^2$L and ours can be viewed as the distribution matching methods, which however still have key differences. In terms of SRe$^2$L, it learns the condensed images by matching the mean and standard deviation of the normalization layer, i.e., matching batch normalization statistics. Instead, we learn the condensed images by matching the real dataset and synthetic dataset with energy distance, which is calculated by the sample level distance. That is to say, SRe$^2$L and ours are complementary. One takes into account the sample level, while the other considers the statistical level when learning images. As we discussed in the revision, ours is flexible and it thus can serve as a regularization term of SRe$^2$L. Specifically, in recover stage of SRe$^2$L, we can assimilate the energy distance between real dataset and synthetic dataset into its loss function to optimize the condensed images. We report the experimental results in revision. The results show that when assimilating ours into  $SRe^2L$, we can get the better performance about the condensed images. It thus proves the flexibility and effectiveness of ours.
>
> Response to W2: Thanks for your valuable suggestion! We thank the reviewer for the insightful suggestion regarding an ablation of the first term in the energy distance (ED) loss. We have considered this variant and reported the corresponding performance when using diversity loss and diversity loss +classifier as the variants in our revision. We found that removing the matching part between real dataset and synthetic dataset usually leads to training instability and degraded performance. This is because that the matching term guides the synthetic samples toward the real data distribution. When discarding the matching term, the resultant loss only pushes the synthetic samples to move away from each other in each class (via the repulsion term), without providing direction toward the target distribution. In other words, the optimization becomes unconstrained in the feature space, making it difficult to converge. This observation is consistent with the theoretical role of the ED structure: the first term enforces global alignment (distribution matching), while the second term promote sample diversity.
>
>
> Response to W3: Thanks for your insightful suggestion! We additional reported the experimental results of DM due to its generalization ability. We ignored other baselines since it might be difficult to adapt them to other modalities. We can find that ours achieves better performance than DM, proving the effectiveness of ours in other modalities.
>
>
> Response to W4: Thanks for your suggestion! Following your suggestion, we have described that how to combine ours with M3D in our revision; please see Sec. 5.2 for more details.

---

### Decision · Action_Editor_u9Vp · 2025-07-06

**Recommendation:** Accept as is

**Additional Comments:**

I recommend accepting this paper based on the evidence provided and the authors' comprehensive revisions during the review process.

Under TMLR's framework, the key question is whether the authors' claims are supported by evidence (not whether those claims represent subjectively significant or novel advances). The authors claim modest computational advantages and competitive performance without bi-level optimization. Both of which are clearly supported by their experimental evidence. The work meets both of TMLR's criteria: claims are evidenced and there is clear audience interest (confirmed by all reviewers).

A final proofreading pass may be beneficial given the amazing amount of extra content the authors have added (but this does not rise to a minor revision).

I want to thank the authors and reviewers for the discussion and the resulting edits that have improved the paper.

**Audience:**

Yes

**Audience Explanation:**

TMLR's audience would be interested in knowing the findings of this paper. Dataset condensation is a relevant topic for the machine learning community, and the proposed approach offers a practical alternative to computationally expensive bi-level optimization methods.

**Claims And Evidence:**

Yes

**Claims Explanation:**

The claims made in the submission are supported by accurate, convincing and clear evidence. The paper provides:

- Comprehensive experimental validation across multiple datasets (CIFAR-10, CIFAR-100, TinyImageNet)
- Proper ablation studies demonstrating the contribution of each component and examining the computational efficiency of the proposed method
- Integration experiments showing flexibility with other methods (M3D, SRe²L)

The authors' claims about avoiding bi-level optimization while maintaining competitive performance are well-supported by the experimental evidence presented.